Report

# Disentangling genetic and epigenetic determinants of ultrafast adaptation

Arne B Gjuvsland[1],[*] (iD), Enikö Zörgö[1,2], Jeevan KA Samy[1], Simon Stenberg[1], Ibrahim H Demirsoy[2] (iD), Francisco Roque[3], Ewa Maciaszczyk-Dziubinska[4], Magdalena Migocka[4], Elisa Alonso-Perez[2], Martin Zackrisson[2], Robert Wysocki[4], Markus J Tamás[2], Inge Jonassen[3], Stig W Omholt[5] & Jonas Warringer[1,2],[**] (iD)

## Abstract

A major rationale for the advocacy of epigenetically mediated adaptive responses is that they facilitate faster adaptation to environmental challenges. This motivated us to develop a theoretical–experimental framework for disclosing the presence of such adaptation-speeding mechanisms in an experimental evolution setting circumventing the need for pursuing costly mutation–accumulation experiments. To this end, we exposed clonal populations of budding yeast to a whole range of stressors. By growth phenotyping, we found that almost complete adaptation to arsenic emerged after a few mitotic cell divisions without involving any phenotypic plasticity. Causative mutations were identified by deep sequencing of the arsenic-adapted populations and reconstructed for validation. Mutation effects on growth phenotypes, and the associated mutational target sizes were quantified and embedded in data-driven individual-based evolutionary population models. We found that the experimentally observed homogeneity of adaptation speed and heterogeneity of molecular solutions could only be accounted for if the mutation rate had been near estimates of the basal mutation rate. The ultrafast adaptation could be fully explained by extensive positive pleiotropy such that all beneficial mutations dramatically enhanced multiple fitness components in concert. As our approach can be exploited across a range of model organisms exposed to a variety of environmental challenges, it may be used for determining the importance of epigenetic adaptation-speeding mechanisms in general.

**Keywords** adaptation; epigenetics; evolution; modelling; population genetics
**Subject Categories** Evolution; Genome-Scale & Integrative Biology
**Mol Syst Biol. (2016) 12: 892**

## Introduction

The need for an extended evolutionary theory where epigenetic mechanisms have a more prominent explanatory position is a much-debated issue (Laland *et al*, 2014). This discussion has arisen due to a deeper understanding of the epigenetic mechanisms underlying phenotypic plasticity and parental influence (Rando & Verstrepen, 2007; Carone *et al*, 2010; Halfmann & Lindquist, 2010; Daxinger & Whitelaw, 2012). A major rationale for advocating an important role for environmentally guided DNA, RNA, protein and metabolite alterations mediated by epigenetic adaptive mechanisms is that such alterations provide an evolutionary advantage by facilitating faster adaptation (Richards, 2006) to new or recurrent environmental changes. To assess the adaptive importance of epigenetic mechanisms relative to a pure mutation–selection regime for a variety of adaptations in a wide range of organisms is a challenging undertaking, however. Even in those cases where we have identified causative genetic variation underlying a specific adaptation and thus may be tempted to promote a gene-centric explanation, we have to show that epigenetic mechanisms have not acted transiently during the adaptation process to guide gene-based solutions by allowing silenced variation to take effect (Masel & Siegal, 2009; Halfmann *et al*, 2010), altering mutation effect sizes (Laland *et al*, 1999; Plucain *et al*, 2014), or enhancing mutation rates either locally (Molinier *et al*, 2006; MacLean *et al*, 2013) or globally (Roth *et al*, 2006; Zhang & Saier, 2009; Martincorena & Luscombe, 2013) through elevated DNA damage (Ruden *et al*, 2008) or impaired DNA repair (Tu *et al*, 1996; Hoege *et al*, 2002; Moore *et al*, 2014; Supek & Lehner, 2015). Such documentation is arguably beyond reach through studies of natural adaptations, but could conceivably be addressed by artificial selection experiments in the laboratory (Conrad *et al*, 2011; Dettman *et al*, 2012).

State-of-the-art experimental evolution methodology can verify the long-term stability of an adaptation after removal of the

1 Centre for Integrative Genetics (CIGENE), Department of Animal and Aquacultural Sciences, Norwegian University of Life Sciences, Ås, Norway
2 Department of Chemistry and Molecular Biology, University of Gothenburg, Gothenburg, Sweden
3 Computational Biology Unit, University of Bergen, Bergen, Norway
4 Institute of Experimental Biology, University of Wroclaw, Wroclaw, Poland
5 Centre for Biodiversity Dynamics, Department of Biology, NTNU – Norwegian University of Science and Technology, Trondheim, Norway
*Corresponding author. Tel: +47 67232713; E-mail: arne.gjuvsland@nmbu.no
**Corresponding author. Tel: +46 317863961; E-mail: jonas.warringer@cmb.gu.se

selection regime creating it, and reversion of candidate mutations by gene editing can validate their adaptive effect. However, refuting epigenetic adaptive mechanisms reviving silenced mutations or influencing mutation rate demands costly mutation–accumulation experiments that precisely mimic the adaptive regime and can currently only address the issue of general changes in mutation rates (Zhu *et al*, 2014). Thus, unless one can remedy these obstacles, even an experimental evolution framework appears impracticable for determining the adaptive importance of epigenetics.

We hypothesized that a possible route to overcome these obstacles would be to make use of a theoretical–experimental approach involving a data-driven evolutionary population model capable of explaining experimental results as a function of mutation effect sizes, mutation target sizes and mutation rate changes. As very fast adaptations are arguably a natural point of departure to search for evolutionarily important epigenetic adaptive mechanisms, we tested this hypothesis by precisely tracking the adaptation dynamics of budding yeast populations adapting to a panel of environmental challenges and by making use of an individual-based evolutionary population model to explain the fastest adaptive trajectories. We found that ultrafast arsenic adaptation could be fully accounted for by gene-based solutions causing extensive positive pleiotropy between fitness components. And we could only account for the experimentally observed homogeneity of adaptive speed and heterogeneity of molecular solutions if mutation rates were close to empirical estimates of the basal mutation rate. As the introduced theoretical–experimental approach can be exploited across a wide range of model organisms and environments (Long *et al*, 2015), it can tentatively become an instrumental generic tool for illuminating the influence of epigenetics on adaptation.

## Results

### Arsenic adaptation is ultrafast and heritable

To identify growth challenges eliciting ultrafast adaptation, we exposed $n = 4$ independent haploid yeast populations, derived from a single clone, to each of 18 energy-constrained environments over 250 generations (Appendix Table S1). The iterative batch experimental evolution (Appendix Fig S1A) forced adapting populations to cycle through a lag phase, an exponential growth phase and a stationary growth phase. We tracked the associated fitness components—length of lag phase, population doubling time and efficiency of growth—at high accuracy (Fig 1A). The four populations, hereafter termed P1–P4, exposed to arsenic in its most toxic form, As(III), adapted faster than populations exposed to other challenges and went from poor to optimal performance for all three fitness components within just a few mitotic divisions (Fig 1B and C, and Appendix Fig S1B). In the absence of arsenic, the adapted strain performed on par with the founder; thus, fitness component increases were adaptive responses to arsenic and not to other selective pressures (Fig 1B). We found estimates of the number of viable cells (colony-forming units; CFU) to match estimates of the total number of cells in populations and to be unaffected by the presence of As(III) (Appendix Fig S1C). The three estimated fitness components therefore reflected the time to the first cell division, cell

division time and the energy efficiency of cells, and together they should capture total fitness well.

The extraordinarily fast As(III) adaptations could conceivably be due to a single rare adaptive mutation standing at substantial frequency in the shared founder population rather than *de novo* mutations. To account for this possibility, we repeated the arsenic adaptations in four new populations, hereafter termed P5–P8, that were initiated from distinct founder populations derived by clonal expansion from four different single cells (see Materials and Methods). As the original As(III) adapting populations (P1–P4), populations P5–P8 showed the ultrafast and near-deterministic adaptive leaps. With the higher sampling density these were detectable already after 10 generations (Fig EV1). The probability of adaptive variants standing in all P5–P8 populations was estimated to be $3.9 \times 10^{-5}$ (see Materials and Methods).

We tested the possibility of the ultrafast adaptation being directly due to non-genetic mechanisms by releasing each of the four adapted populations P1–P4 from selection for 75 generations. All populations retained their extreme As(III) tolerance (Fig 1D). This excludes a plain phenotypic plasticity mechanism at the level of the individual cell. Moreover, we are not aware of any reports of trans-generational epigenetic inheritance of fitness over 75 generations, making it a quite unlikely explanation.

### Ultrafast arsenic adaptation is driven by *FPS1*, *ASK10* and *ACR3* mutations

The genetic nature of As(III) adaptation motivated us to sequence the end point populations P1–P4 by SoLID sequencing to identify *de novo* single nucleotide polymorphisms (SNPs) and copy number variations (CNVs) rising to high frequencies (Appendix Tables S2 and S3). To identify and validate causative mutations, 35 of the top candidates were individually reconstructed in founder cells and fitness components were recorded (Fig 2A). About 75% of final adaptive gains in each of the four populations could be explained by a single population specific mutation. Adaptation in P2, P3 and P4 was predominantly due to a premature stop codon in *FPS1* (P4; encoding the aquaglyceroporin through which As(III) enters the cell; Wysocki *et al*, 2001), a duplication of *ACR3* (P2; encoding the As(III) exporter; Wysocki *et al*, 1997) and a loss-of-function SNP in *ASK10* (P3; a positive regulator of Fps1; Lee *et al*, 2013), respectively (Fig EV2). The mutations were neutral (Acr3, Ask10) or negative (Fps1) in absence of As(III), excluding that they were driven to high frequencies by other selection pressures (Appendix Fig S2). The P1 population harboured two medium-frequency *FPS1* non-synonymous mutations (A410S and F413L) that resisted reconstruction, but were predicted to impair Fps1 function by amino acid conservation over Fps1 orthologs. The P1 mutations occurred in different haplotypes (reads), affecting close to the entire population (Appendix Table S2).

Conceivably, early As(III) adaptation could have been epigenetic in origin and only later assimilated as *FPS1*, *ASK10* and *ACR3* mutations into the genome (Pigliucci *et al*, 2006). We therefore sequenced the P1–P4 populations throughout their adaptive trajectories by Illumina sequencing, tracking the frequency change of these causative mutations over time (Fig 2B). *FPS1*, *ASK10* and *ACR3* mutations were all at undetectable frequencies in the founder population, emerged and rose in frequency early and were practically

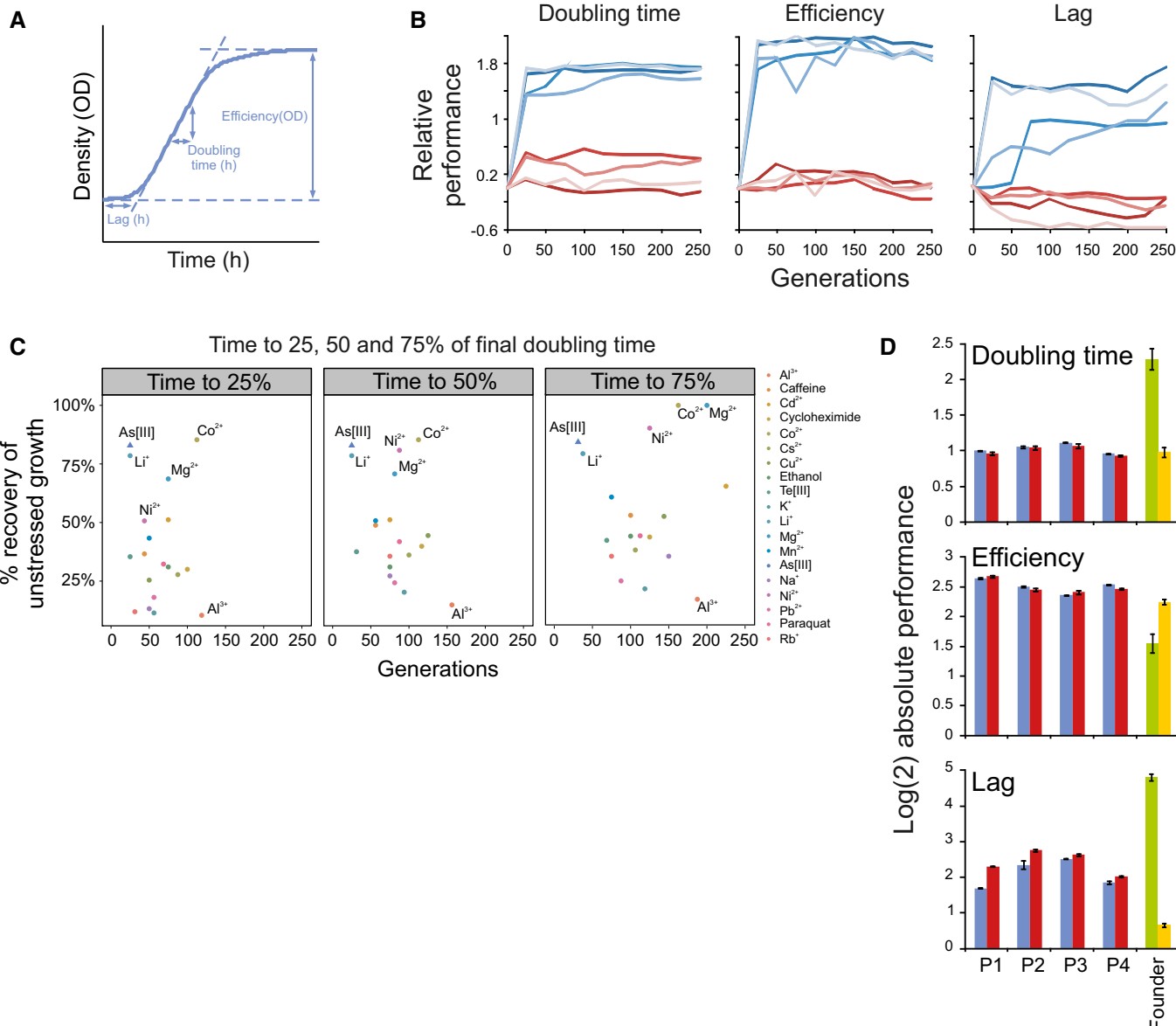

**Figure 1. Arsenic adaptation is ultrafast and heritable.**

A Schematic illustration of the extraction of the fitness components: length of lag phase (h), growth rate (doubling time, h) and growth efficiency (total change in population size, OD). Absolute fitness components were log(2)-transformed. When comparisons across experimental plates were performed, absolute log(2) values were first normalized to the corresponding mean value of many founder populations in randomized positions on the same plate, producing relative fitness components (see Materials and Methods). A positive relative performance equals good growth.

B Fitness components of As(III) adapting populations (*n* = 2) relative the founder (*n* = 4). Blue = 5 mM As(III). Red = no As(III). Colour = populations P1–P4.

C Mean adaptation speed under 18 selection pressures (*n* = 4 independent populations) for doubling time. A monotone function was fitted to each adaptation curve using least squares and the function *isoreg* in *R* (version 2.15.3). Two measures of adaptive speed were extracted from the function: (*x*-axis) the number of generations required to reach 25, 50 and 75% of the final doubling time (*t* = 250 generations) and (*y*-axis) the fraction (%) of the initial gap to the founder doubling time in optimal environments (no stress added) that was recovered at these time points. Colour indicates challenge.

D Absolute log(2) fitness components of arsenic-adapted (*t* = 250 generations) populations in 5 mM As(III), before (blue bars) and after (red bars) a 75 generations release from selection. Green bars: founder in 5 mM As(III). Yellow bars: founder without arsenic. Error bars represent SEM (*n* = 2).

Source data are available online for this figure.

fixated before 100 generations. The rise in frequency of the *ACR3* duplication was slightly delayed relative the adaptive progression in P2 (compare: Figs 1B and 2B), which may be due to sequencing or negative selection against the large duplication during sequencing preparations. No other SNPs from the original end point sequencing were confidently called at earlier time points, and only a few previously undetected SNPs were discovered (Appendix Fig S3). Of these, only three were predicted to affect protein function (see Materials

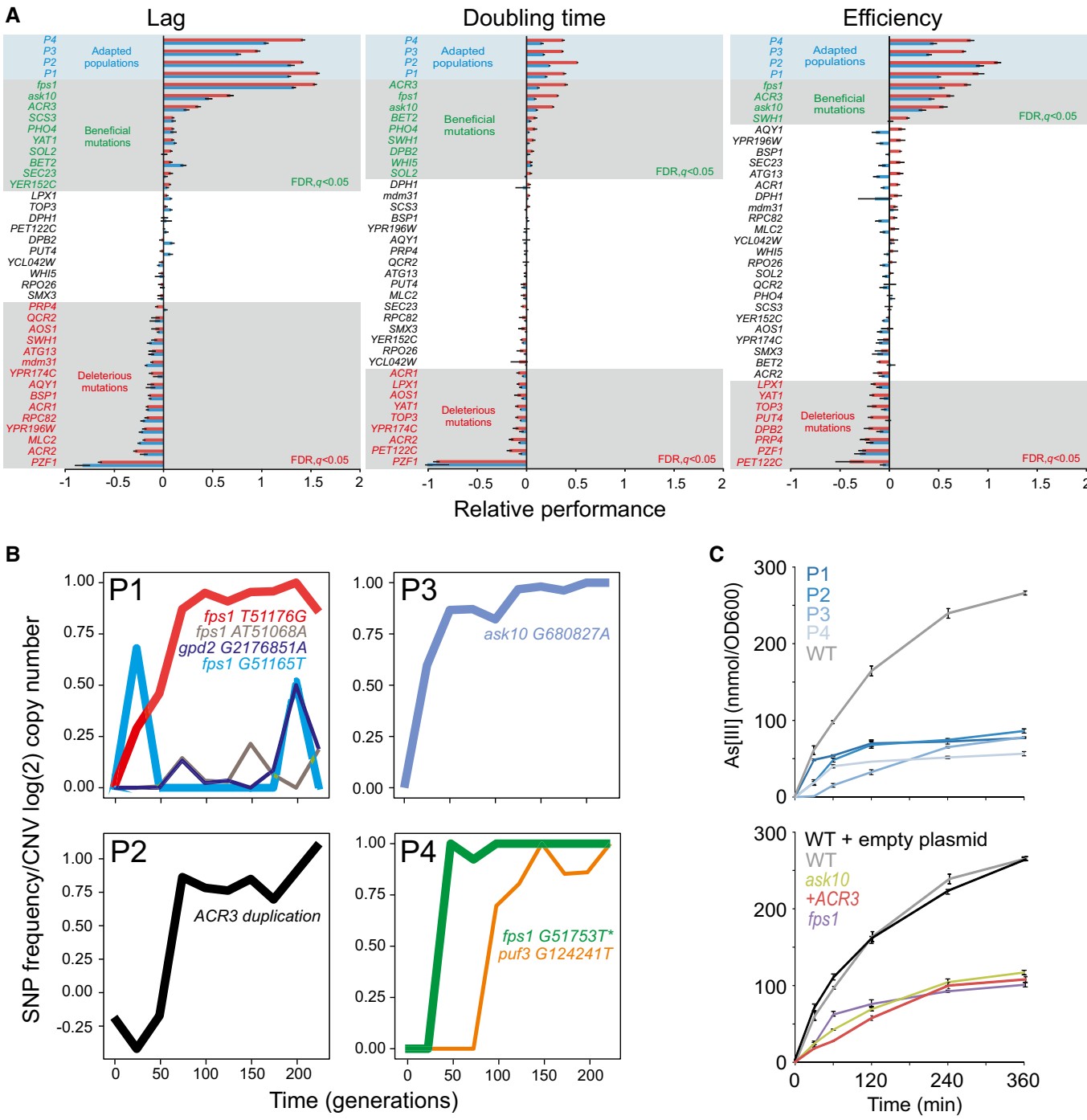

**Figure 2. Ultrafast arsenic adaptation is due to rapid fixation of positively pleiotropic FPS1, ASK10 and ACR3 mutations.**

A Candidate driver mutations (gene duplications or SNP; see Materials and Methods) were reconstructed individually in a WT background and their growth in presence of As(III) evaluated. Mean (*n* = 2) fitness components in 3 and 5 mM As(III) (blue and red bars, respectively) relative to the founder (*n* = 20) are shown. Grey field = significant (FDR, *q* < 0.05) effects at 5 mM As(III). Blue field = As(III) adapted populations. Error bars represent SEM.

B Adapting P1–P4 populations were sampled at every 25th generation and deep sequenced. The frequency of confidently called mutations (total read depth of > 100, frequency of > 10% in ≥ 2 time points and snpEff effect = "Moderate" or "High") predicted to affect protein function (SIFT < 0.05) is shown. Fps1 G51165T failed to pass the quality filter in the re-sequencing but is shown for completeness. For the *ACR3* containing duplicated region in P2, the grand copy number mean of all the segments within the duplicated region (chr. XVI 880799–944600) is shown. Colours indicate mutations. Bold line = causative mutations. *reconstructed *FPS1* mutation.

C Arsenic accumulation inside cells. Top panel: Arsenic-adapted populations (*t* = 250 generations). Bottom panel: As(III) causative mutations individually reconstructed in WT backgrounds. Error bars = SD (*n* = 2).

Source data are available online for this figure.

and Methods): a previously undetected, low-frequency frameshift in *FPS1* (P1), a late emerging and low-frequency *GPD2* mutation (P1) and a late emerging mutation in *PUF3* (P4) (Fig 2B). Neither Gpd2, the minor isoform of the glycerol dehydrogenase, nor Puf3, involved in mitochondrial function and mRNA stability, is known to be linked to As(III) metabolism. Overall, although we cannot completely exclude very small transient contributions from epigenetics or from variants in other genes, loss-of-function mutations in *FPS1* and *ASK10* and duplications of *ACR3* emerged as the dominant proximal causes of ultrafast As(III) adaptation.

As the identified causative mutations implied adaptation to be mediated by As(III) exclusion, we followed the accumulation of arsenic inside cells. We found it to be delayed and stabilized at a lower final level in all adapted populations (Fig 2C, top panel). The reduced intracellular arsenic levels were almost completely accounted for by the reconstructed *fps1*, *ask10* and *ACR3* mutations (Fig 2C, bottom panel). Thus, by reducing the intracellular concentration of arsenic, these mutations affected all three fitness components through the same mechanism: As(III) exclusion.

## Ultrafast arsenic adaptation is due to positive pleiotropy between fitness components

Given these experimental results, we assessed quantitatively to what extent the observed adaptive trajectories could be accounted for by plain neo-Darwinian mechanisms. We employed an evolutionary population model based on individual cells that combined population genetics and population dynamics through genotype–phenotype maps with parameters describing mutation rates and effect sizes. As each of the three reconstructed critical mutations had a large positive effect on all three fitness components, we first evaluated whether positive pleiotropy was needed to account for the ultrafast adaptation. To this end, we simulated the experimental set-up while varying mutation parameters and using three types of genotype–phenotype maps: mutations affecting only cell division time (population doubling time), mutations affecting both cell division time and the time to the first cell division (population lag time) in the same direction (positive pleiotropy) and mutations affecting both doubling time and lag but with independently sampled effects (random assignment of positive and negative pleiotropy). Efficiency was not taken into account because it may not confer fitness benefits in an energy-restricted regime (MacLean, 2008) and would require a much more complex model with weak empirical backing.

The parameter set giving the fastest adaptation for the doubling time-only model clearly failed to approach the ultrafast adaptation of arsenic adaptations (Fig 3A) despite: (i) having overall mutation rates 5× those reported in yeast (Lynch *et al*, 2008), (ii) 65% of mutations being non-neutral and 17% of mutations being beneficial (both values 5× the numbers reported by (Hall *et al*, 2008) and (iii) a mean selection coefficient of 0.15 (2.5× the level reported by Joseph & Hall, 2004). Overall, populations adapted dramatically faster in the models with pleiotropy between fitness components than in the models without fitness component pleiotropy (Figs 3B and EV3A and B). Populations exclusively experiencing positive pleiotropy adapted only moderately faster than populations experiencing both positive and negative pleiotropy. Thus, positive pleiotropy between fitness components can indeed accelerate adaptation drastically, and the presence of negative pleiotropy only moderately

limits this acceleration. The benefits of positive pleiotropy were similar for slow- and fast-adapting populations (Appendix Fig S4). With positive pleiotropy included, the fastest scenarios approached the observed arsenic adaptations (Fig 3B).

Next, we used the empirical lag and doubling time values for reconstructed mutations and simulated the fate of the *de novo FPS1*, *ASK10* and *ACR3* mutations in competition assays, starting with a single-mutant cell at the start of the first batch cycle in an otherwise homogenous founder population (Fig 3C). We simulated mutants with only the doubling time effect of the reconstructed mutation, only the lag effect and both the doubling time and the lag effect. Factoring in both the doubling time and lag effect strongly reduced the risk of losing the *FPS1* mutation due to chance (loss in 14 of 25 doubling time scenarios vs. 0 of 25 doubling time and lag scenarios) and accelerated fixation of remaining mutations. The positively pleiotropic doubling time and lag effects combined into a very large selection coefficient ($s = 0.64$; Appendix Fig S5), driving the mutation to fixation in ~25 generations. Thus, the predicted adaptive performance approached that of the experimentally ultrafast arsenic adaptations, without even taking efficiency into account. The selection coefficients for *ASK10* and *ACR3* were somewhat lower ($s = 0.41$ and $s = 0.36$ respectively, Appendix Fig S5), causing a slightly longer fixation time (Fig 3C). The additive effects of doubling time and lag changes on fitness are shown analytically in the Materials and Methods section.

## Ultrafast arsenic adaptation occurs at near-basal mutation rates

Even though the competition simulations show that the focal causal mutations can account for the ultrafast adaptation, these results rest on that mutations emerge early. Given that epigenetic mechanisms can direct DNA damage and DNA repair (Molinier *et al*, 2006; Roth *et al*, 2006; Zhang & Saier, 2009; MacLean *et al*, 2013; Martincorena & Luscombe, 2013) to elevate mutation rates, this early emergence may in principle be epigenetically facilitated. To resolve this issue, we defined the beneficial mutation target set to contain *ACR3* duplications and loss-of-function mutations in *FPS1* and *ASK10*, the mutational targets of the latter being premature stop codons and changes in strongly conserved amino acids (SIFT < 0.06; Appendix Fig S6A and B). We simulated populations experiencing basal as well as elevated (3×, 5× and 10×) point mutation and duplication rates and tracked the frequencies of all beneficial mutations using lag and rate values equalling those empirically observed. At basal mutation rates, the founder genotype went extinct within 25 generations in the fastest evolving populations (Fig 3D—upper left panel). While the predicted large variations in adaptive speed between populations at a basal mutation rate (Fig 3D—upper left panel; compare time to vertical black line) are a possible scenario, it produces a distribution of simulated adaptations from which we would be unlikely to draw the four nearly deterministic ultrafast adaptations observed in the experimental data (Fig 1B and Appendix Fig S2A). A mutation rate closer to the upper bound of empirical estimates of the basal mutation rate (3×) (Lynch, 2006; Lang & Murray, 2008; Lynch *et al*, 2008) increased the homogeneity in adaptive speed considerably, while allowing heterogeneity in adaptive solutions. In this case, the founder genotype went extinct in 35 generations in the median population (Fig 3D, upper right panel). Mutation rates (> 5×) above empirical estimates of the basal

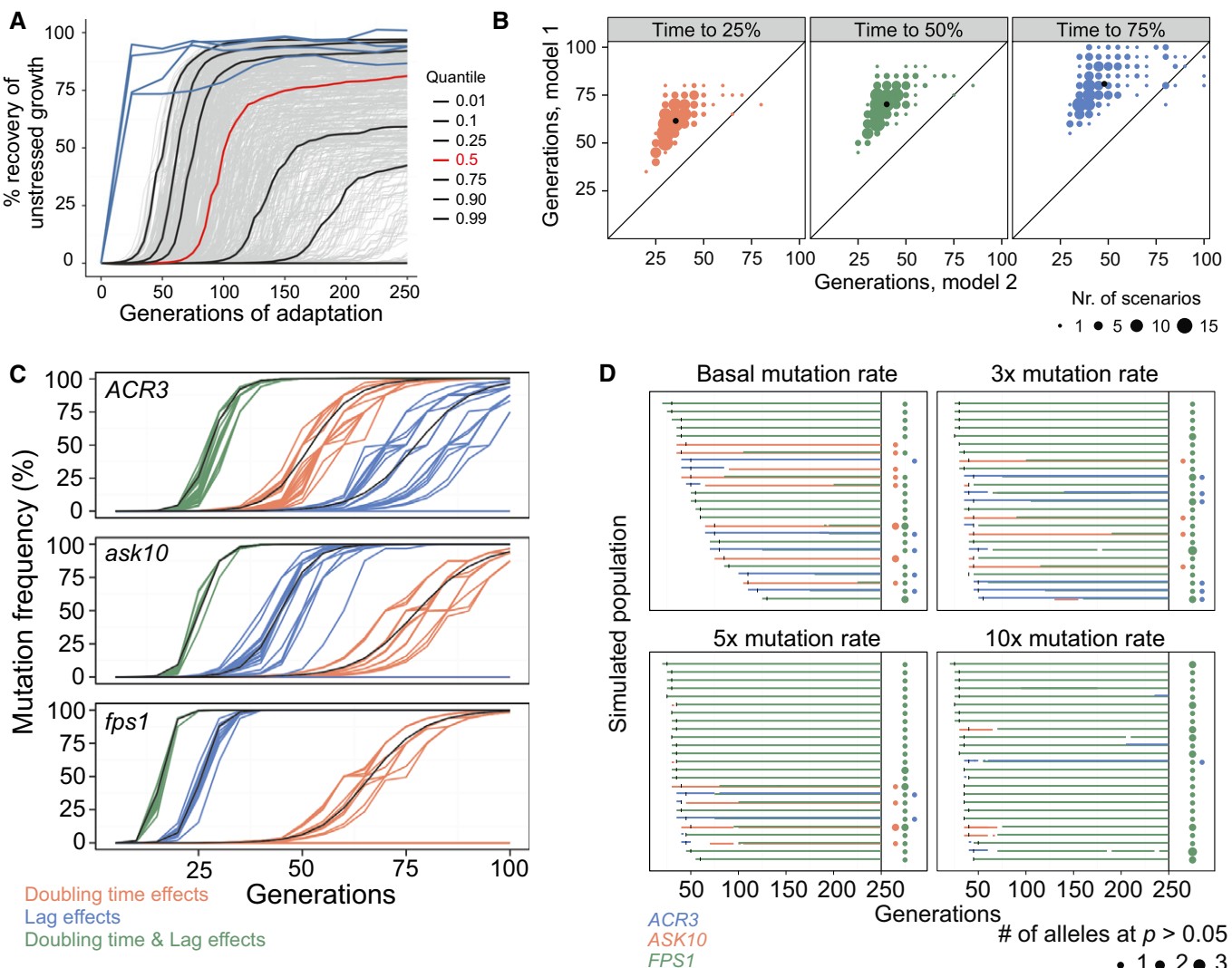

**Figure 3. Ultrafast arsenic adaptation is explained by plain neo-Darwinian mechanisms.**

A, B   Simulating populations (*n* = 500) with different mutation parameters adapting to arsenic, in an individual cell-based model. Each cell has a cell division time (population doubling time) and a time to the first cell division (lag). Cell division times were recorded every 5th cell division, and population means were expressed as the fraction of founder growth w/o arsenic recovered. (A) Mutations affect only cell division time. Grey lines = 500 adapting populations. Black lines = quantiles corresponding to the fastest 1, 10, 25, 75, 90 and 99% of populations. Red line = median population. Blue line = empirical arsenic populations, P1–P4. (B) Contrasting simulations with mutations only affecting cell division time (model 1, M1, *y*-axis) or both cell division time and the time to the first cell division with the same effect size and direction (model 2, M2, *x*-axis). The number of cell divisions required to recover 25% (left panel), 50% (middle panel) and 75% (right panel) of founder growth w/o arsenic is shown. The 227 fastest scenarios, with 75% recovery in ≤ 100 cell divisions, are displayed. Dot size = number of populations. Black dot = median population.

C   Simulated competition in arsenic between WT and *ACR3* (top panel), *ASK10* (middle panel), and *FPS1* (lower panel) mutation-carrying cells, respectively. Mutations emerge in a single cell at *t* = 0 in WT population and have empirical effects on doubling time and/or growth lag. Coloured lines = model with individual cells and genetic drift. Black lines = deterministic model based on subpopulations of cells.

D   Arsenic adaptation of 25 simulated populations in an individual cell-based model, given basal, 3×, 5× and 10× mutation rates. Beneficial mutations in *FPS1* (green), *ASK10* (red) and *ACR3* (blue) occur randomly at estimated frequencies and have empirical lag and doubling time effects. Left hand side: Before *t* = 250 generations. Horizontal bars = mutated alleles exist at (sum) *P* > 50%. Vertical black lines = WT allele goes extinct. Right hand side: At *t* = 250 generations. Dot presence = WT allele extinct. Dot colour = gene name. Dot size = number of mutated alleles at *P* > 5%.

mutation rate gave results that were incompatible with the experimental data, as the superior *FPS1* mutations were then consistently fixed, leaving no room for *ACR3*- and *ASK10*-based solutions (Fig 3D, lower panels). The simulations therefore suggested a mutation rate between 1× and 3× at loci under selection and excluded substantially higher mutation rates.

The confinement of the mutation rate needed to explain the co-appearance of homogeneity in adaptation speed and heterogeneity in adaptive solutions is in line with existing theory of adaptation dynamics in asexual populations (Sniegowski & Gerrish, 2010) (Appendix Fig S7). With a mutation rate of ~1–3×, our dataset falls in the *strong selection, strong mutation* regime, where a handful of

mutations compete for fixation. Higher mutation rate brings us into the *strong selection, weak mutation* regime, where *FPS1* with its high selection coefficient is the preferred solution. At lower mutation rates, we move towards the *weak selection, strong mutation* regime, where rare mutation events lead to large variation in adaptation speed.

Notably, in contrast to our experimental observations, the model predicted that *FPS1* mutations would eventually emerge and fixate in populations where *ACR3* or *ASK10* were already fixated (Fig 3D). The reconstructed *FPS1*, *ASK10* and *ACR3* mutations explain most of the fitness gains seen in P1–P4, but not all of it. Unidentified causal mutations accounted for ~25% of total fitness gains, and this fraction is larger in *ASK10* and *ACR3* populations than in *FPS1* populations. We may therefore have underestimated the late fitness of *ACR3* and *ASK10* containing clones relative to *FPS1* containing clones. Indeed, when we mimicked this possibility by letting the adaptive gains of *ASK10* and *ACR3* mutations approach that of *FPS1*, *FPS1* was less prone to fixate in *ASK10/ACR3* backgrounds (Appendix Fig S8).

There are no robust experimental means to estimate local mutation rates at loci under selection. However, loss-of-function mutation rates can be estimated for marker genes (Lang & Murray, 2008) and used as somewhat crude proxies. We therefore tested the model prediction that *FPS1*, *ASK10* and *ACR3* mutation rates are at the most moderately elevated in cells adapting to As(III) by measuring the loss-of-function mutation rate for *CAN1* at its native locus (Fig EV4). Both in non-adapted (WT) and adapted (*FPS1* mutated) genotypes, we found a slightly elevated (1.5–1.6×) loss-of-function mutation rate at the *CAN1* locus during As(III) exposure. These estimates do not allow precise conclusions on loss-of-function mutation rates at *FPS1* and *ASK10* loci, and emphatically not for the duplication rate at the *ACR3* locus. However, they exclude a dramatic elevation of the general point mutation rate during arsenic exposure, in agreement with model predictions.

## Discussion

As all large effect mutations driving As(III) adaptation enhanced multiple fitness components, positive pleiotropy appeared as the main reason for the observed ultrafast adaptation, a conclusion confirmed by the theoretical modelling. The underlying reason being that the large effect mutations all contributed to fast exclusion of As(III), preventing As(III) from accumulating inside cells and thereby from delaying the time to the first cell division, delaying cell division and increasing the energy costs of homeostasis maintenance.

Mutations inactivated Fps1, the main entrance pathway for As(III) or its critical activator, Ask10, or duplicated the major extrusion system, Acr3 (Fig EV2), in agreement with that Fps1 and Acr3 are the most critical contributors to intracellular As(III) accumulation and toxicity (Wysocki *et al*, 1997, 2001; Talemi *et al*, 2014). The ultrafast adaptation observed for As(III) was not shared by the other selective pressures evaluated. This may be explained by the nature of the elements used. First, several toxic metals share properties and structures with critical metals. For instance, toxic $Na^+$, $Li^+$, $Rb^+$ and $Cs^+$ are similar to $K^+$, which plays many important physiological roles and exert their toxicity partially by substituting $K^+$ in the

cell (Cyert & Philpott, 2013). Likewise, the toxicity of $Cd^{2+}$ can in part be attributed to its similarity with essential elements, such as $Zn^{2+}$ and $Ca^{2+}$ (Wysocki & Tamas, 2010). These toxic metals often use the same transport systems to enter or leave the cell as the essential metal and a simple exclusion strategy, as for As(III), will not be possible. Second, some elements are critical at low concentrations but toxic at elevated levels, for example $Cu^{2+}$. For these elements, solutions in the form of exclusion are likely to be severely constrained by their essentiality (Wysocki & Tamas, 2010; Cyert & Philpott, 2013). In contrast, As(III) is always toxic and has no known function in cells. Third, an element may "hijack" an important cellular process for mediating its toxicity (e.g. the sulphate assimilation pathway is central to Te(IV) toxicity) (Ottosson *et al*, 2010). Such a mode of toxic action is likely to prevent fast adaptation. Fourth, exclusion of toxic elements may be mediated by multiple pathways. In these cases, exclusion may be constrained by redundancy such that no single mutation has a large effect. Finally, even when mutations at a locus have large effects, the locus may be too small for mutations to be encountered frequently. Adaptation to rapamycin is, for example, dominated by large effect mutations in the rapamycin binding protein Fpr1 (Lorenz & Heitman, 1995), but the gene is a tiny 345 bp and the expected waiting time for loss-of-function mutations is therefore long.

There are currently two approaches to measure and model fitness in experimental populations (Barrick & Lenski, 2013). The standard approach measures the fitness of individual genotypes as their frequency change over time in competition assays (Gresham *et al*, 2008; Lang *et al*, 2011). This is simplified if each genome in a population is barcoded before the onset of selection with a unique sequence tag (Levy *et al*, 2015; Venkataram *et al*, 2016), allowing very accurate estimation of the fitness distribution of standing and de novo mutations for use as a model input. Given that change in fitness of the population is also exactly measured and a suitable modelling framework in place, such approaches are certainly useful for understanding the speed of adaptation. So far, however, these approaches have focused on steady-state adaptation where a population has evolved in a constant environment for a long time, with selection acting only on doubling time (Kosheleva & Desai, 2013; Rice *et al*, 2015).

We employed the alternative approach: break fitness in batch-to-batch experiments down into its components, both experimentally and theoretically. This approach certainly comes with caveats attached. It is not always clear that the estimated and modelled fitness components—here cell division time and time to the first cell division—fully capture fitness. In experimental microbial populations, death rates may not be negligible and it is debatable whether efficient use of resources, as reflected in the final growth yield of a population, is a selectable trait or not (MacLean, 2008; Ibstedt *et al*, 2015). Furthermore, to estimate fitness components, mutations must be reconstructed or reversed and the fitness component of individual genotypes must be estimated. This is laborious, in particular if interactions between mutations and between individuals (Moore *et al*, 2013) are to be measured. Here, we considered evolutionary scenarios of very fast adaptation, where single mutations drive adaptation and rapidly rise to fixation, without measurable death occurring. In such scenarios, the caveats above are lesser concerns. Under slow, absent or negative adaptation, clonal interference, positive epistasis, cell–cell interactions and death may all be substantial.

In such evolutionary scenarios, more complex models may be needed.

A marked benefit of breaking fitness down, and connecting it to genotypes via the intervening phenotypic layers, is the possibility to identify the causal factors underlying particular patterns of adaptation. This is illustrated by our discovery that positive pleiotropy between fitness components is the driving force of the observed ultrafast adaptation. To understand adaptation dynamics at an even deeper level, both experimentation and modelling must be extended to molecular phenotypes. For example, by connecting the time to the first cell division and the cell division time to the biochemical and network properties of As(III) metabolism (Talemi *et al*, 2014), a complete and formalized understanding of the causes of ultrafast As(III) adaptation could potentially be obtained.

In conclusion, our results show that even ultrafast adaptation can be achieved based on purely genetic, de novo solutions, without invoking either direct or indirect action of epigenetics (Lenski & Mittler, 1993; Brisson, 2003; Galhardo *et al*, 2007; Ram & Hadany, 2012). Proof by example provides no grounds for rejecting the hypothesis that transgenerational epigenetic mechanisms mediating fast organismal adaptation can be evolutionarily relevant. However, as adaptation speed is a frequent argument for why adaptation by transgenerational epigenetic mechanisms would be favoured by selection and become widespread in nature, our result is a reminder of the forcefulness of plain neo-Darwinian adaptation mechanisms. But the major instrumental value of our study is that it provides a framework of generic worth across a range of experimentally evolvable organisms and environments to systematically assess how important epigenetic mechanisms are for achieving fast adaptation.

## Materials and Methods

### Strains and medium

Haploid, asexual BY4741 cells (*MATa*; *his3*Δ1; *leu2*Δ0; *met15*Δ0; *ura3*Δ0) (Brachmann *et al*, 1998), stored at −80°C in 20% glycerol, were used as WT and to initiate founder populations. Gene duplication events were mimicked by transforming WT cells with centromeric *URA3* and *KANMX4* plasmids from the MoBY collection (Ho *et al*, 2009). Each plasmid contained a single gene (BY4741 alleles). Point mutations were individually reconstructed in BY4741 backgrounds using *in vivo* site-specific mutagenesis, as described (Stuckey *et al*, 2011). Strains were cultivated in synthetically complete (SC) medium containing: 0.14% yeast nitrogen base (YNB, CYN2210, ForMedium), 0.50% ammonium sulphate, 0.077% complete supplement mixture (CSM, DCS0019, ForMedium), 2.0% (w/v) glucose and pH buffered to 5.8 with 1.0% (w/v) succinic acid and 0.6% (w/v) NaOH. Except for glucose, all required nutrients were present in excess. Where indicated, the medium was supplemented with 3 or 5 mM $NaAsO_2$ (As(III), Sigma-Aldrich) or other environmental challenges as described in Appendix Table S1.

### Experimental evolution

Except for the four follow-up arsenic adapting populations, P5–P8, reported in Fig EV1, all experimental evolutions were initiated from a single founder population. The founder population was constructed by clonal colony expansion up to an estimated 1 million cells, from a single cell, on SC agar medium (as above, +2% (w/v) agar) with no added stress. The colony was dissolved in liquid SC medium to create the founder population, the optical density was measured, and an average of $10^5$ cells were randomly drawn by pipetting of 5 μl of cell suspension into experimental wells to initiate each adaptation. The follow-up experiment of arsenic adapting populations, P5–P8, was initiated identically, except that each of the four populations was initiated from four different founder populations. These were clonally expanded from four distinct cells, up to a population size of ~$3 \times 10^7$. Assuming that adaptive mutations during the clonal expansion from a single cell are Poisson distributed, with normal mutation rates and the mutation target sizes reported in Appendix Fig S6, the probability that a single P5–P8 population housed one or more standing adaptive variants is ~0.08. The probability that all of P5–P8 housed one or more standing adaptive variant at experiment start is ~$3.9 \times 10^{-5}$. Experimental evolutions were performed in a batch-to-batch mode in flat-bottom 96-well micro-titre plates containing SC complete medium supplemented with stress factors (Appendix Table S1). To reduce the risk of cross-contamination, every second well was left empty, such that all pairs of populations were separated by empty wells. No indication of cross-contamination between As(III) populations was found in the sequence data. Except for the follow-up As(III) adapting populations P5–P8, populations were propagated over 50 batch-to-batch cycles as 175 μl, non-shaken cultures maintained at 30°C. The follow-up As(III) adapting populations P5–P8 were propagated over 20 cycles. In all cycles, populations were cultivated well into stationary phase. The cultivation length corresponding to ~120-h cultures over the first 10 cycles, ~96 h in cycles 10–30 and ~72 h in cycles 30–50. Stationary phase population sizes corresponded to on average $N = 3.5 \times 10^6$ cells, with the largest deviation corresponding to half that size. To initiate each new cycle, 5 μl of re-suspended and randomly drawn stationary phase cell cultures, corresponding to an average of $N = 10^5$ cells, was multi-pipetted into fresh medium. Each batch cycle corresponded to ~5 population size doublings. The adaptation schema thus progressed over ~250 population doublings (~100 doublings for follow-up As(III) adapting populations P5–P8). Except for the follow-up arsenic adaptations, P5–P8, 50 μl of each population was sampled at the end of every $5^{th}$ cycle, pipetted into 100 μl of 30% glycerol and stored as a frozen fossil record at −80°C. For populations P5–P8, sampling was instead performed at every batch cycle.

### Fitness component extraction

To estimate fitness components, frozen samples were first thawed and re-suspended. 10 μl was pipetted into random wells in 100-well honeycomb plates, each well containing 350 μl of liquid SC medium. Populations were pre-cultivated without shaking at 30°C for 72 h until well into stationary phase. Following re-suspension, 10 μl of each pre-culture was randomly sampled and transferred to 100-well honeycomb plates, containing 350 μl of liquid SC medium supplemented by relevant stress factors. Populations were cultivated in Bioscreen C (Growth Curves Oy, Finland) instruments for 72 h at 30°C and at maximum horizontal shaking for 60 s every other minute (Warringer & Blomberg, 2003; Warringer *et al*, 2003).

Optical density (turbidity) was recorded every 20 min using a wide-band (420–580 nm) filter. Stochastic noise was removed by light smoothing of the raw data, the background light scattering was subtracted, and optical densities were transformed into population size estimates using an empirically based calibration (Fernandez-Ricaud *et al*, 2016). From population size growth curves, population doubling times, length of the lag phase and population growth efficiency (total gain in population size) were extracted (Fernandez-Ricaud *et al*, 2016). Population growth parameters were log(2)-transformed to better adhere to normal distribution assumptions. When comparisons across plates were made, log(2) estimates were first normalized to the corresponding mean of 4–20 WT (founder) controls distributed in fixed but randomized positions. For doubling time and lag, the relative growth measures equalled: mean of log(2) WT estimates – log(2) experimental estimate. For the population growth efficiency, the relative growth measure equalled: log(2) experimental estimate – mean of log(2) WT estimates. Positive values thus always indicate adaptation. The normalization accounts for systematic bias between plates, instruments and batches. Finally, a mean was formed across replicates.

### Viable and total cell counts

We cultivated populations of founder (BY4741) and adapted (the reconstructed *FPS1* mutation) genotypes in 100-well honeycomb plates in 350 µl of SC medium (as above) with and without 5 mM As(III) until OD = 1.00 ($n = 12$). To count viable cells, we plated diluted cells on solid (1.5% agar) medium with no added arsenic and counted the number of colony-forming units (CFU) and multiplied by the dilution factor. No significant difference in CFU was observed between presence and absence of 5 mM As(III) for either founder or *FPS1* cells. We counted total cells in two ways, both by passing sonicated (30 s; to dissolve cell aggregates) and diluted samples through a flow cytometer (BD FACSaria, BD Biosciences, US) at a known flow rate, counting passage events and multiplying by the dilution factor and by direct counting of cells in a hemocytometer (Bürker counting chamber, Knittel Gläser).

### As(III) accumulation

Exponentially growing cells (in 150 ml of complete SD medium, with or without uracil) were exposed to 1 mM As(III) and sampled at indicated time points. Cells were washed (2×) in ice-cold water and centrifuged. Cell pellets were re-suspended in water, boiled (10 min) and centrifuged to collect the supernatant. The As(III) content of each sample was measured ($n = 2$) using a flame atomic absorption spectrometer (3300, Perkin Elmer).

### Fluctuation assay mutation rate estimation

The *CAN1* fluctuation assay was performed as described (Lang & Murray, 2008). Single streaked WT and reconstructed *FPS1* colonies were isolated on solid SC medium and inoculated and cultivated overnight in SC medium. Cultures were diluted to a fixed cell density and distributed into the wells of four (WT and *FPS1*, with and without As(III) 96-well plates, each containing 25 µl of SC medium. Wells were sealed to prevent evaporation and cross-contamination and cultivated for 3 days at 30°C, without shaking.

We counted cells in three random wells for each plate with a hemo-cytometer (as above), using the mean as an estimate of cell count per wells in each sample. Discarding wells in the outer frame, we plated the remaining 57 independent cultures on SC agar medium lacking arginine but containing 0.6 g/l of *L*-canavanine sulphate (Sigma-Aldrich). After 3 days at 30°C, we estimated the fraction, $P_0$, of plated patches without any canavanine-resistant colonies (i.e. without colonies carrying *CAN1* loss-of-function mutations). The *CAN1* loss-of-function mutation rate, $\mu$, (mutations per *CAN1* locus per cell division) was then estimated as: $\mu = -\ln(P_0)/N$.

### Sequencing and sequence analysis

To sequence founder and As(III) adapting populations, frozen samples were thawed and re-suspended and 10 µl was pipetted into 100 µl of SC medium with weak (2 or 3 mM) As(III) selection, minimizing allele frequency change. Populations were cultivated into stationary phase. DNA from end point populations was extracted, prepared and sequenced using ABI 5,500 × l SOLiD™ sequencing, completely according to industrial standards. Quality controlled reads were aligned to the yeast reference genome (sacCer3) using ABI's BioScope v1.3. SNPs and indels were called using SAMtools mpileup (Li *et al*, 2009), filtering for regions > 5× mean coverage. CNVs were called using a sliding window approach, with window size: 300 bp and step length: 300 bp. A CNV was conservatively called using CNV seq (Xie & Tammi, 2009) if the log(2) coverage ratio founder/adapted population) exceeded 0.5 or fell short of −0.5. For re-sequencing of earlier time points, DNA was extracted using Epicentre MasterPure Yeast DNA Purification kit, with an added lyticase digestion step. Libraries were prepared using the Illumina Nextera XT enzymatic kit. Paired-end sequencing was performed on a HiSeq 2500, according to industrial standards. Reads were quality-trimmed (Phred score cut-off of 25). Nextera transposase sequences were removed using Trim Galore (v.0.3.8). Reads were mapped to the S288C reference genome (R64-1-1_20110203) using BWA MEM (v.0.7.7-r441). PCR duplicates were removed post-mapping using Picard tools (v.1.109). Samples were sequenced 4–6×, and libraries from the same sample were merged, again using Picard tools. To avoid false variant calls as a result of misalignment around indels, base alignment quality scores were calculated using SAMtools (v.0.1.18 [r982:295]) (Li, 2011). Variants were called using FreeBayes (v0.9.14-8-g1618f7e), treating all sequenced time points from each population as a cohort and filtered in three steps. First, we removed variants with a quality score < 20. Second, we removed variants standing in our founder population ($P > 10\%$ in at least two libraries of founder cultivated in absence of As(III)) relative the reference genome. Third, we removed variants standing in the founder after DNA preparative cultivation (3 mM As(III)) ($P > 0.2$). Variants were annotated using snpEff (v.3.6c). To only retain mutations likely to affect protein function (Fig 2B), we filtered for "Moderate" and "High" snpEff calls and further for non-synonymous SNPs with a SIFT (Sorting Intolerant From Tolerant; Ng & Henikoff, 2003) score of < 0.05. Appendix Fig S3 shows the results without the protein function filtering. Nextera enzyme digestion is biased, making CNV calling inaccurate. We therefore only estimated the copy number of the *ACR3* region, calling the duplicated segment using a sliding window with a size and step

length = 100 bp. Reads with mapping quality < 1 were discarded. Segments were calculated with the circular binary segmentation algorithm using DNAcopy (v.1.40.0) in R.

## Deterministic modelling of competition assays

As a basis for the individual-based stochastic model, we developed a deterministic model of competition assays using simple subpopulation growth curves. This deterministic model was used to (i) analytically solve (black lines) fates of novel mutations (Fig 3C) and (ii) to analytically convert mutation effects on doubling time and lag into a selection coefficient (Appendix Fig S5). The model describes a competition assay between two subpopulations, WT ($P_1$) and mutated ($P_2$), in a batch cycle set-up mimicking the experimental framework (Appendix Fig S1A). Within each subpopulation (index $i = 1, 2$) individuals share genotype and genetically determined fitness component values. We let $N_i(t)$ be the population size of population $P_i$ at time $t$. Batch cycles start with $N_i(0)$ individuals in $P_i$. The total bottleneck population size is $N = N_1(0) + N_2(0)$. No net growth occurs in the time period until time $\lambda_i$ (lag period). Thereafter, growth is exponential with doubling time $\tau_i$, giving:

$$N_i(t) = N_i(0)2^{\frac{t-\lambda_i}{\tau_i}} \tag{1}$$

A batch cycle ends at time $t_{end}$ after $M$ population doublings such that: $2^M N = N_1(t_{end}) + N_2(t_{end})$.

### *Simulations underlying black lines in Fig 3C*
Parameter settings: $N = 10^5$, $M = 5$ and lag ($\lambda_{WT} = 804.8$, $\lambda_{fps1} = 276.6$, $\lambda_{ask10} = 503.3$ $\lambda_{acr3} = 630.4$ min) and doubling times ($\tau_{WT} = 162.3$, $\tau_{fps1} = 130.2$, $\tau_{ask10} = 134.6$ $\tau_{acr3} = 122.8$ min) based on mean empirical values. We assumed *ACR3* duplication effects and the marginal effect of the plasmid to be multiplicative. We simulated competition assays over 20 cycles starting from a single-mutant cell ($N_2(0) = 1$) in the first cycle (Fig 3C black lines). We recorded the frequency ratio, $r(t) = N_2(t)/N_1(t)$ of mutant over WT genotypes. We computed mutation selection coefficients, $s$, by regressing $ln(r(t))$ on the number of (P1) generations (Appendix Fig S5A). Finally, we estimated the sensitivity of these selection coefficient estimates to potential measurement error in lag and doubling time (Appendix Fig S5B).

### *Analytic results on selection coefficients*
With a simplifying assumption on $t_{end}$, we also derived analytical expressions for the joint contribution of lag and doubling time effects on selection coefficients. We assumed $\lambda_2 \leq \lambda_1$ and computed $r(t)$ for a single cycle starting at $t = 0$ and ending at $t_{end} = \lambda_1 + M\tau_1$ where P1 has doubled $M$ times.

$$
\begin{aligned}
r(t) &= r_0, & 0 \leq t \leq \lambda_2 \\
r(t) &= r_0 2^{\frac{t-\lambda_2}{\tau_2}}, & \lambda_2 < t \leq \lambda_1 \\
r(t) &= r_0 2^{\frac{t-\lambda_2}{\tau_2} - \frac{t-\lambda_1}{\tau_1}}, & \lambda_1 < t
\end{aligned}
\tag{2}
$$

The selection coefficient $s$ expressed as the per generation slope of $ln(r(t))$ becomes:

$$
\begin{aligned}
s &= [ln(r(M\tau_1 - \lambda_1)) - ln(r(0))]/M \\
s &= \left[ ln\left( r_0 2^{\frac{(M\tau_1 + \lambda_1) - \lambda_2}{\tau_2} - \frac{(M\tau_1 + \lambda_1) - \lambda_1}{\tau_1}} - ln(r_0) \right) \right]/M \\
s &= \left[ ln\left( 2^{\frac{M\tau_1 + \lambda_1 - \lambda_2}{\tau_2} - M} \right) \right]/M \\
s &= ln(2)\left[ \frac{M\tau_1 + \lambda_1 - \lambda_2}{\tau_2} - M \right]/M \\
s &= ln(2)\left[ \left( \frac{\tau_1}{\tau_2} - 1 \right) + \frac{\lambda_1 - \lambda_2}{M\tau_2} \right]
\end{aligned}
\tag{3}
$$

When mutations only affect doubling time ($\lambda_2 = \lambda_1$), equation 3 simplifies to

$$s = ln(2)\left( \frac{\tau_1}{\tau_2} - 1 \right) \tag{4}$$

This is equivalent to equation 3.2 in (Chevin, 2011). When mutations only affect lag time ($\tau_1 = \tau_2$), equation 3 simplifies to

$$s = ln(2)\left( \frac{\lambda_1 - \lambda_2}{M\tau_2} \right) \tag{5}$$

Thus, the selection coefficient due to a difference in lag time is reduced when the number of mitotic divisions between bottlenecks or the doubling time increases. Furthermore, for the parameter values for our reconstructed mutation, the lag and doubling time effects on selection coefficients are close to additive.

## Individual-based model of batch experimental evolution

To account for the combined effects of random mutation events, random subsampling of cells, clonal interference and epistasis, we mimicked the experimental framework (Appendix Fig S1A) in an individual-based model. Each cell has its individual genotype that determines the time to the first cell division and its cell division time. Cells with identical genotypes divide at the same time. Parameters, similar to those for the deterministic model above, describe population size at the start of each batch cycle ($N$), average number of mitotic divisions before serial transfer ($M$) and total number of batch cycles. When the total population size reaches $2^M N$ cells, $N$ cells are subsampled randomly to found the next cycle. The model assumed no cell death, that all effects on cell division rate and lag are genetic, no meiosis or ploidy change and no interactions between cells. We found no evidence of cell death, no evidence of ploidy change in sequence data and meiosis is inactivated by deletion of the mating type-switching gene. The model was used for three sets of simulations of increasing complexity:

### *Individual-based competition assays*
We simulated the competition assays studied with subpopulation growth curves ($N = 10^5$, $M = 5$, 20 batch cycles, genotypic lag and doubling time parameters as above) in Fig 3C (coloured lines). Competition assays were initiated with a single-mutant (*fps1*, *ask10* or *ACR3*) cell in a founder population and no other mutations emerging. Frequency trajectories are stochastic due to the random sampling of individuals. In the extreme case, random sampling leads to loss-of-mutant lines.

## Simulating experimental evolution with mutations in *FPS1*, *ASK10*, *ACR3*

We simulated experimental evolution starting from a founder population accumulating mutations in *FPS1* (loss-of-function), *ASK10* (loss-of-function) and *ACR3* (duplication). The basal duplication rate of *ACR3* was set to $3 \times 10^{-7}$ duplications per division. Beneficial mutational target sizes for *FPS1* and *ASK10* were computed by downloading the SIFT yeast database (http://sift-db.bii.a-star.edu.sg/public/Saccharomyces_cerevisiae/EF4.74/) and extracting all possible stop gain base changes and non-synonymous mutations with attached SIFT scores, for *FPS1* and *ASK10* (Appendix Fig S6). In Fig 3D, beneficial mutation was considered as all mutations with a SIFT score < 0.06, corresponding to the observed driver mutation with the highest SIFT score. Mutational target sizes were multiplied with a global mutation rate estimate (Lynch *et al*, 2008) of $0.33 \times 10^{-9}$ mutations/bp/division. Empirical lag and doubling time values were used (above), assuming complete negative epistasis. We ran four sets of simulations ($n = 25$) over 50 cycles with $N = 10^5$ and $M = 5$, varying the mutation rate from basal to 3×, 5× and 10× the basal rate.

## Simulating adaptation across the realistic range of mutation parameter values

We simulated the experimental framework ($N = 10^5$, $M = 5$, 50 batch cycles) starting from a clonal wild-type (WT) population with empirical arsenic WT lag and doubling time values ($L_{wt} = 805$ min, $T_{wt} = 162$ min) adapting towards wild-type performance in absence of arsenic ($L_{wt,N} = 271$ min, $T_{wt,N} = 126$ min). Mutation events were sampled after each cell division with parameters being the overall mutation rate, the proportion of mutations affecting fitness, the proportion of fitness-affecting mutations that are beneficial, and the distribution of selection coefficients for non-neutral mutations. Following (Joseph & Hall, 2004), the selection coefficient $s_r$ for a given mutation $m$ was sampled from a gamma distribution with shape $\alpha$ and scale $\beta$. The selection coefficients in Joseph and Hall (2004) are given by the ratio of exponential growth rates of the mutant and wild-type strain, respectively, and following (Chevin, 2011) we compensated for the overestimation (factor $ln(2)$) of the per generation selection coefficients in equations 3–5. Sampled selection coefficients, $s = s_r\, ln(2)$, were inserted into equations 3–5 and rearranged to provide the mutation induced change in doubling or lag time from the WT. For doubling time, the equation becomes

$$\Delta T_{wt}^{m} = \frac{T_{wt}s_r}{s_r + 1} \tag{6}$$

$\Delta T_{wt}^{m}$ is the change in doubling time when mutation $m$ emerges as the first mutation in a WT cell with doubling time $T_{wt}$. In the very rare cases where $\Delta T_{wt}^{m}$ exceeded $T_{wt} - T_{wt,N}$, the value was truncated. Negative epistasis was implemented in the form of diminishing return of positive mutations. If the mutation $m$ emerges in a cell with genotype $G$ and doubling time $T_G$, the resulting change in doubling time was modelled as

$$\Delta T_{G}^{m} = \frac{T_{G} - T_{wt,N}}{T_{wt} - T_{wt,N}} \Delta T_{wt}^{m} \tag{7}$$

The capping of extreme effects and diminishing return of consecutive positive mutations means that adapting cells asymptotically approach WT growth in absence of arsenic. All empirical populations followed this behaviour (Figs 1A and EV1), and there is strong experimental support for a diminishing return of positive mutations (Chou *et al*, 2011; Khan *et al*, 2011). We simulated models where mutations affected only cell division time (population doubling time, M1), cell division time (population doubling time), and time to the first cell division (population lag time) with the same effect sign and size (M2) and cell division time (population doubling time), and time to the first cell division (population lag time) with random effect sign and magnitudes (M3) using 500 mutation parameter sets based on empirical values. These corresponded to full-factorial combinations of the mutation rate ($\frac{\mu}{5}, \frac{\mu}{3}, \mu, 3\mu, 5\mu$, where the base mutation rate $\mu = 0.33 \times 10^{-9}$ mutations/bp/division; Lynch *et al*, 2008), the fraction of fitness-affecting mutations ($\frac{y}{5}, \frac{y}{3}, y, 3y, 5y$, where $y = 0.034$ (Hall *et al*, 2008), the fraction of fitness-affecting mutations that are beneficial ($\frac{z}{5}, \frac{z}{3}, z, 3z, 5z$, where $z = 0.13$; Hall *et al*, 2008) and the scale ($\beta = 13.3, 20, 27.35, 33, 40, 47$ min) of the effect size distribution. Thirty-three minutes corresponded to the reported empirical value (Joseph & Hall, 2004). The shape of the effect size distribution was kept constant at $\alpha = 2$. We recorded population averages for doubling times at the end of each batch cycle.

## Data availability

Models are available as Code EV1 and can also be downloaded from https://bitbucket.org/ajkarloss/yeast_sim. The SOLiD sequencing data are accessible at EBI (http://www.ebi.ac.uk/ena/data/view/PRJEB17740) with accession number PRJEB17740. The Illumina sequencing data are accessible at NCBI (https://www.ncbi.nlm.nih.gov/sra?term=SRP092403) with accession number SRP092403.

**Expanded View** for this article is available online.

## Acknowledgements

We thank Payam Ghiaci for help with DNA extraction. Financial support by the Polish National Science Centre (grant number 2012/07/B/NZ1/02804) to RW, from the Swedish Research Council (grant numbers 325-2014-6547 and 621-2014-4605) and from the Carl Tryggers Foundation (grant number CTS 12:521) to JW, and from the Research Council of Norway (grant numbers 178901/V30 and 222364/F20) is acknowledged.

## Author contributions

ABG, SWO and JW conceived, designed and coordinated the study. EZ, IHD, SS and MZ performed and analysed evolution experiments. EA-P performed mutation rate experiments. EM-D, MM, RW and MJT designed, performed and analysed arsenic accumulation. SS, FR and IJ analysed sequence data. ABG and JKAS designed, performed and analysed simulations. ABG, SWO and JW wrote the paper, with input from all other authors.

## Conflict of interest

The authors declare that they have no conflict of interest.

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
