## [Review Process File · Molecular Systems Biology]

Disentangling genetic and epigenetic determinants of ultrafast adaptation

Arne B. Gjuvslund, Ms. Enikő Zörgö, Mr. Jeevan Samy, Mr. Simon Stenberg, Mr. Ibrahim Demirsoy, Francisco Roque, Ewa Maciaszczyk-Dziubinska, Magdalena Migocka, Elisa Alonso-Perez, Mr. Martin Zackrisson, Robert Wysocki, Markus Tamás, Inge Jonassen, Stig Omholt and Jonas Warringer

Corresponding authors: Jonas Warringer, Göteborg University

Arne B. Gjuvslund, Norwegian University of Life Sciences

Review timeline:

Submission date:	15 March 2016
Editorial Decision:	13 May 2016
Revision received:	30 September 2016
Editorial Decision:	27 October 2016
Revision received:	11 November 2016
Accepted:	16 November 2016

Transaction Report:

1st Editorial Decision

13 May 2016

Thank you again for submitting your work to Molecular Systems Biology. We have now heard back from the three referees who agreed to evaluate your manuscript. As you will see from the reports below, while referee #3 is not positive, the two other referees find the topic of your study of potential interest. They raise, however, concerns on your work, which should be convincingly addressed in a revision of this work.

The major concerns raised by the reviewers refer to the following points:

- the mutation rate should be measured in rich medium and high arsenic conditions
- the frequency of the adaptive mutation and its progression should be monitored (and its absence initially) in the very early population.
- the methods and results should be presented in a much clearer and structured way.

REFeree REPORTS

Reviewer #1:

There have been several recent claims that microorganisms adapt rapidly to new conditions by

epigenetic adaptation rather than through selection acting on genetic mutations. In this manuscript Warringer and co-workers present evidence that one such 'ultrafast' adaptation - the adaptation of yeast to arsenic - is genetic not epigenetic. This is an important and provocative result - most of the claims of epigenetic adaptation have been without a demonstrated mechanism and without formally ruling out adaptation via selection acting on mutations. Here they take the correct scientific approach and test whether selection on mutations can explain the ultrafast adaptation rather than simply claiming an 'epigenetic' mechanism. They present quite convincing evidence that it is the underlying mechanism. The combination of experiment and modelling is elegant.

Suggestions:

I think the paper is missing one crucial piece of evidence for their model - that the adaptive mutations are actually present in the very early populations and that they increase in frequency with the expected timescale. They identify the mutations and confirm their adaptive benefit, but they don't directly demonstrate that they are the early cause of adaptation rather than some putative epigenetic adaptation followed by a mutation fixing. Rather only the plausibility of this is shown through the modelling.

Why is ultrafast adaptation observed for arsenic but not for other conditions? It would be interesting for the authors to discuss their ideas about this at the end of the manuscript. Some of the other traits have potentially large mutation target sizes.

The mutation rate in the models doesn't quite fit the measured mutation rate, therefore not quite ruling out an increase in mutation rate in stress conditions (?).

Reviewer #2:

This paper studies the molecular mechanisms underlying unusually fast evolutionary adaptation in the eukaryotic model *Saccharomyces cerevisiae*. In short, the authors report that populations of yeast cells can adapt quite quickly (around 25 generations) to toxic arsenic concentrations. They show that the adaptation is heritable and subsequently go on to investigate if the fast adaptation can be explained through purely genetic changes (i.e. mutations), or whether epigenetic changes may also be involved. To answer this question, they developed a computational model that shows that if the mutation rates are slightly (3x) higher than the previously published basal levels, the swift adaptation can be entirely explained through genetic changes, without the need for epigenetic mechanisms.

Overall, this paper tries to address a very interesting question, and the approach taken is quite innovative. However, I think that the paper is a bit premature and would greatly benefit from a few (relatively simple albeit labor-intensive) controls.

1. I think it is absolutely necessary (and not very difficult) to measure the mutation rates in rich medium and in high-arsenic conditions (using classic fluctuation assays). In this way, the authors can prove that the mutation rates in arsenic are really what their model predicts. Without such accurate measurements, the whole study seems completely useless because it makes it impossible to show whether mutations alone can really explain the fast adaptation. I honestly suspect that the conclusions will remain, as it seems likely that the mutation rate is higher than expected under these conditions - but we obviously need to know for sure!

2. As a control, I think it is essential to also study a control case where adaptation happens at an average pace. Does the model make accurate predictions here too?

3. I am not sure whether the model accurately models what is happening in the population. Specifically, I wonder whether the model also takes into account the possibility of unadapted cells dying (and disappearing from the population)? If this really happens, it may change the predictions of the selection coefficient and mutation rates. Obviously, the most elegant way to test the accuracy of the model would be to experimentally validate it using the lineage-tracking strategy developed by Gavin Sherlock's team (doi: 10.1038/nature14279.).

4. I would like to see more profiles of adaptation in (much) smaller starting populations, to rule out

the possibility of standing variation (adaptive mutations that are already present in the founder population) contributing to the adaptation.

5. The model used for investigating the effect of epigenetics on fast emergence of beneficial mutations is not completely clear to me. As far as I understand, the beneficial mutation targets are limited to the genes identified. Ideally, should this not also include (unknown) targets? Can this be discussed a bit more in the text?

6. In the first paragraph of the results section, the authors say that the populations exposed to arsenic go "from poor to optimal performance" and they refer to figure 1A. However, that figure seems to indicate that the lag phase becomes longer as generations go by, which suggests that the populations are not (yet) optimal, and that there may be a tradeoff. Many figures represent the doubling time, efficiency and lag time of the evolved/evolving populations compared to founder. This implies that the ratio of the evolved populations over the founder is used. In this case, adaptive changes in fitness would occur when the ratio for doubling time and lag time becomes negative; or when the ratio for efficiency becomes positive. However, all relative performances in Fig 1 increase, suggesting that the absolute doubling time increases and the lag time increased in the evolved populations. Please discuss. Moreover, I find the evolution experiments poorly described. What was the culture volume and system? Which was the initial OD? Did the cultures reach stationary phase before each transfer? These factors affect the selection regime and thus also the outcome of the experiment. The same questions apply to the experiments used to extract the fitness components.

7. While the main text is well written, the figure legends and methods section are below standards. Specifically, the authors should describe in more detail how the experimental evolution was performed (temperature, were the cells grown in flasks or tubes or plates, in what volume, etc.). Other specific sentences that were unclear are listed in the minor comments.

Minor questions and suggestions

1. While growth rates and doubling times are two sides of the same coin ($GR = \ln(2)/DT$), they should not be used interchangeably. When presenting doubling times, this should not be labeled as rates. This is especially confusing in Fig 2b, which shows that the 'rate' of the founder is higher in the presence of arsenic (compared to a benign condition), suggesting that the founder is growing at a higher growth rate in the presence of arsenic. It is also difficult to understand why a $\log(2)$ transformation is necessary when presenting these absolute measures of doubling time, efficiency and lag.

2. Second paragraph of Results section: which are the "repeated arsenic adaptations P5-P8"? Populations P1-P4 and P5-P8 are never explained.

3. Page 6, "positive pleiotropy vastly accelerated adaptation": according to the figures, model 2 and model 3 don't seem to be so vastly different.

4. Last paragraph in page 6: the authors say they assume that the mutations emerge early. If they would sequence the intermediate time-points, they could know for sure when each of the mutations emerge.

5. Page 7, "adaptive speed did not accord with experimental data": Where can we see this?

Reviewer #3:

The paper by Gjuvsland et al attempts to argue that the pattern of evolution observed in a adaptation of yeast to high arsenic is fully compatible with the absence of epigenetic mechanism that increase the rate of mutation. In this way, the authors, claim they can disentangle the genetic and epigenetic mechanisms.

I do not believe this paper is ready for publication in any journal, and certainly not in Mol. Syst. Biology. The description of the experiment is obscure with most results in Supp. Information. The first paragraph of the Results starts with citing Table S1 and Fig. S1A, S1B, C, and D. It is not at all obvious how the fitnesses and clonal interference patterns have been characterized and from what I could surmise with great difficulty consisted of just sequencing a few clones. The paper brings very little new thinking to the field as well. Desai and Fisher 2007 and subsequent work is an attempt to explain the data using simple models and population sequencing and even more so barcoding experiments of Levy et al provide much better resolution to answer these questions.

Finally, the fit of poor data into a simple model is a poor way to argue that epigenetic mechanisms could not have been involved.

1st Revision - authors' response

30 September 2016

Editor's comments: Thank you again for submitting your work to Molecular Systems Biology. We have now heard back from the three referees who agreed to evaluate your manuscript. As you will see from the reports below, while referee #3 is not positive, the two other referees find the topic of your study of potential interest. They raise, however, concerns on your work, which should be convincingly addressed in a revision of this work. The major concerns raised by the reviewers refer to the following points:

- the mutation rate should be measured in rich medium and high arsenic conditions
- the frequency of the adaptive mutation and its progression should be monitored (and its absence initially) in the very early population.
- the methods and results should be presented in a much clearer and structured way.

If you feel you can satisfactorily deal with these points and those listed by the referees, you may wish to submit a revised version of your manuscript. Please attach a covering letter giving details of the way in which you have handled each of the points raised by the referees. A revised manuscript will be once again subject to review and you probably understand that we can give you no guarantee at this stage that the eventual outcome will be favorable.

Authors: We now address these and other points raised by the reviewers. The point-by-point response follows below. In short, we have:

- Thoroughly revised and extended the methods description and figure legends, aiming for enhanced clarity and structure (comment 9, 10, 16, 18)
- Added text and figures corresponding to three new experiments - on allele frequency change (comment 1, 14), mutation rate (comment 4) and viability assays (comment 6) that supports the reported conclusions.
- Substantially expanded the Results and Discussion to illuminate issues brought up by the reviewers (comments 2, 3, 5, 7, 8, 19)
- Done minor text modifications as requested by the reviewers (11, 12, 13, 15)
- Elevated two supplementary figures to text figures (comment 17)
- Explained why we think the somewhat negative perspective brought forward by the third reviewer to be incorrect (comment 20)

We have included (copy-paste) all text changes into the responses below.

Reviewer #1: There have been several recent claims that microorganisms adapt rapidly to new conditions by epigenetic adaptation rather than through selection acting on genetic mutations. In this manuscript Warringer and co-workers present evidence that one such 'ultrafast' adaptation - the adaptation of yeast to arsenic - is genetic not epigenetic. This is an important and provocative result - most of the claims of epigenetic adaptation have been without a demonstrated mechanism and without formally ruling out adaptation via selection acting on mutations. Here they take the correct scientific approach and test whether selection on mutations can explain the ultrafast adaptation rather than simply claiming an 'epigenetic' mechanism. They present quite convincing evidence that it is the underlying mechanism. The combination of experiment and modelling is elegant.

Suggestions:

1. I think the paper is missing one crucial piece of evidence for their model - that the adaptive mutations are actually present in the very early populations and that they increase in frequency with the expected timescale. They identify the mutations and confirm their adaptive benefit, but they don't directly demonstrate that they are the early cause of adaptation rather than some putative

epigenetic adaptation followed by a mutation fixing. Rather only the plausibility of this is shown through the modelling.

Author response: We have now deep sequenced all populations at nine additional time points during adaptation. We show that the frequencies of the suggested adaptive mutations follow the dynamics of adaptation remarkably well. The relevant Results section now reads:

“Conceivably, early As(III) adaptation could have been epigenetic in origin and only later assimilated as *FPS1*, *ASK10* and *ACR3* mutations into the genome (Pigliucci et al, 2006). We therefore sequenced the P1-P4 populations throughout their adaptive trajectories by Illumina sequencing, tracking the frequency change of these causative mutations over time (Fig 2B). *FPS1*, *ASK10* and *ACR3* mutations were all at undetectable frequencies in the founder population, emerged and rose in frequency early and were practically fixated before 100 generations. No other point mutations confidently called by both sequencing techniques followed the same behaviour. The rise in frequency of the *ACR3* duplication was slightly delayed relative the adaptive progression in P2 (compare Fig 1B and 2B). This may be due to sequencing or negative selection against any of the duplicated genes in the segment during sequencing preparations. No other SNPs from the original end point sequencing were confidently called at earlier time points and only a few previously undetected SNPs were discovered (Fig S3). Of these, only three were predicted to affect protein function (see Methods): a previously undetected, low frequency (P1) frameshift in *FPS1*, a late emerging (P1) and low frequency *GPD2* mutation and a late emerging (P4) mutation in *PUF3* (Fig 2B). Neither *Gpd2*, the minor isoform of the glycerol dehydrogenase, or *Puf3*, involved in mitochondrial function and mRNA stability, are known to be linked to As(III) metabolism. Overall, although we cannot completely exclude very small contributions from transient epigenetics or from variants in other genes, loss-of-function mutations in *FPS1* and *ASK10*, and duplications of *ACR3* clearly emerged as the dominant proximal causes of ultrafast As(III) adaptation.”

2. Why is ultrafast adaptation observed for arsenic but not for other conditions? It would be interesting for the authors to discuss their ideas about this at the end of the manuscript. Some of the other traits have potentially large mutation target sizes.

Authors: We now discuss this in the Discussion which reads:

“Mutations inactivated *Fps1*, the main entrance pathway for As(III) or its critical activator, *Ask10*, or duplicated the major extrusion system, *Acr3* (Fig S2B), in agreement with that *Fps1* and *Acr3* are the most critical contributors to intracellular As(III) accumulation and toxicity (Talemi et al, 2014; Wysocki et al, 1997; Wysocki et al, 2001). The ultrafast adaptation observed for As(III) was not shared by the other selective pressures evaluated. This may be explained by the nature of the elements used. First, several toxic metals share properties and structures with critical metals. For instance, toxic Na^+ , Li^+ , Rb^+ , and Cs^+ are similar to K^+ , which plays many important physiological roles, and exert their toxicity partially by substituting K^+ in the cell (Cyert & Philpott, 2013). Likewise, the toxicity of Cd^{2+} can in part be attributed to its similarity with essential elements, such as Zn^{2+} and Ca^{2+} (Wysocki & Tamas, 2010). These toxic metals often use the same transport systems to enter or leave the cell as the essential metal and a simple exclusion strategy, as for As(III), will not be possible. Second, some elements are critical at low concentrations but toxic at elevated levels, e.g. Cu^{2+} . For these elements, solutions in the form of exclusion are likely to be severely constrained by their essentiality (Cyert & Philpott, 2013; Wysocki & Tamas, 2010). In contrast, As(III) is always toxic and has no known function in cells. Third, an element may ‘hijack’ an important cellular process for mediating its toxicity (e.g. the sulfate assimilation pathway is central to Te(IV) toxicity) (Ottosson et al, 2010). Such a mode of toxic action is likely to prevent fast adaptation. Fourth, exclusion of toxic elements may be mediated by multiple pathways. In such cases, exclusion may be constrained by redundancy such that no single mutation has a large effect. Finally, even when mutations at a locus have large effects, the locus may be too small for mutations to be encountered frequently. Adaptation to rapamycin is e.g. dominated by large effect mutations in

the rapamycin binding protein Fpr1 (Lorenz & Heitman, 1995), but the gene is a tiny 345bp and the expected waiting time for loss-of-function mutations is therefore long.”

We also emphasize that a more complete addressing of this issue requires advances beyond this paper:

“To understand adaptation dynamics at an even deeper level, both experimentation and modelling must be extended to molecular phenotypes. For example, by connecting the time to the first cell division and the cell division time to the biochemical and network properties of As(III) metabolism (Talemi et al, 2014), a complete and formalized understanding of the causes of ultrafast As(III) adaptation could be obtained.”

3. The mutation rate in the models doesn't quite fit the measured mutation rate, therefore not quite ruling out an increase in mutation rate in stress conditions (?).

Authors: Our model results rule out a manifold increase in basal mutation rate and points towards mutation rate in the range of 1-3x the basal mutations. By following the mutation rate at a neutral marker loci (see response to reviewer 2, comment 1), we find empirical support for a slight increase in mutation rate (1.5x). We cannot say with complete confidence that this mutation rate well represents those at the adaptive loci. However, it rules out a substantial general increase in mutation rate. Overall, the measured adaptation rate is slightly faster than model predictions. All models are simplifications of reality, and estimates of both mutation rates and fitness component effects do come with measurement error attached. Some discrepancy between models and empirical measures is therefore unavoidable. However, we feel confident in the key conclusion from this section: that substantial increase in the relevant mutation rates is not compatible with the results obtained. We now discuss this more extensively - see the response to reviewer 2, comment 4 and 8.

Reviewer #2: This paper studies the molecular mechanisms underlying unusually fast evolutionary adaptation in the eukaryotic model *Saccharomyces cerevisiae*. In short, the authors report that populations of yeast cells can adapt quite quickly (around 25 generations) to toxic arsenic concentrations. They show that the adaptation is heritable and subsequently go on to investigate if the fast adaptation can be explained through purely genetic changes (i.e. mutations), or whether epigenetic changes may also be involved. To answer this question, they developed a computational model that shows that if the mutation rates are slightly (3x) higher than the previously published basal levels, the swift adaptation can be entirely explained through genetic changes, without the need for epigenetic mechanisms. Overall, this paper tries to address a very interesting question, and the approach taken is quite innovative. However, I think that the paper is a bit premature and would greatly benefit from a few (relatively simple albeit labor-intensive) controls.

4. I think it is absolutely necessary (and not very difficult) to measure the mutation rates in rich medium and in high-arsenic conditions (using classic fluctuation assays). In this way, the authors can prove that the mutation rates in arsenic are really what their model predicts. Without such accurate measurements, the whole study seems completely useless because it makes it impossible to show whether mutations alone can really explain the fast adaptation. I honestly suspect that the conclusions will remain, as it seems likely that the mutation rate is higher than expected under these conditions - but we obviously need to know for sure!

Author response: To address this issue, we performed a *CANI* fluctuation assay for two genotypes, WT and reconstructed *FPSI* mutants. The assay shows a modest 1.5x fold increase in the *CANI* loss-of-function mutation rate during As(III) exposure. The increase in the two genotypes was near identical. The extrapolation of mutation rates from marker loci to other loci is certainly precarious and some caution in interpretation is prudent. However, the measures well match model predictions: 1x-3x fold increase in the mutation rate at the causative loci and the results reassuringly validates that that at least the general mutation rate is unlikely to be substantially increased during As(III) exposure.

The Results reads:

“There are no robust experimental means to estimate local mutation rates at loci under selection. However, loss-of-function mutation rates can be estimated for marker genes (Lang & Murray, 2008) and used as somewhat crude proxies. We therefore tested the model prediction that *FPS1*, *ASK10* and *ACR3* mutation rates are at the most moderately elevated in cells adapting to As(III) by measuring the loss-of-function mutation rate for *CAN1* at its native locus (Fig S9). Both in non-adapted (WT) and adapted (*FPS1* mutated) genotypes, we found a slightly elevated (1.5-1.6x) loss-of-function mutation rate at the *CAN1* locus during As(III) exposure. These estimates do not allow precise conclusions on loss-of-function mutation rates at *FPS1* and *ASK10* loci, and emphatically not for the duplication rate at the *ACR3* locus. However, they exclude a dramatic elevation of the general point mutation rate during arsenic exposure, in agreement with model predictions.”

5. As a control, I think it is essential to also study a control case where adaptation happens at an average pace. Does the model make accurate predictions here too?

Author response: Accurate predictions across the whole span of adaptation and extinction dynamics would indeed be important to argue that the model has general predictive capacity. However, demonstrating such accuracy is a huge and very complex task that we believe extends much beyond the boundaries of this paper. We have re-written the Discussion to make it clear that we do not claim that the model, in the here implemented form, has such general predictive capacity. This section reads:

“Here, we considered evolutionary scenarios of very fast adaptation, where single mutations drive adaptation and rapidly rise to fixation, without measurable death occurring. In such scenarios, the caveats above are lesser concerns. Under slow, absent or negative adaptation, clonal interference, positive epistasis, cell-cell interactions and death may all be substantial. In such evolutionary scenarios, more complex models may be needed.”

6. I am not sure whether the model accurately models what is happening in the population. Specifically, I wonder whether the model also takes into account the possibility of unadapted cells dying (and disappearing from the population)? If this really happens, it may change the predictions of the selection coefficient and mutation rates. Obviously, the most elegant way to test the accuracy of the model would be to experimentally validate it using the lineage-tracking strategy developed by Gavin Sherlock's team (doi: 10.1038/nature14279.).

Author response: We counted viable and total cells in absence and presence of As(III) and found no measurable death to occur. Viability was indistinguishable, with and without As(III). Thus, the confounding factor is not relevant for the reported results. This is now shown and commented on in the Results. We also highlight that in other evolutionary scenarios, the model may need to be extended with death rate estimates.

“We found estimates of the number of viable cells (colony forming units; CFU) to match estimates of the total number of cells in populations, and to be unaffected by the presence of As(III) (Fig S1C). The three estimated fitness components therefore reflected the time to the first cell division, cell division time and the energy efficiency of cells, and together they should capture total fitness well”

“In experimental microbial populations, death rates may not be negligible and it is debatable whether efficient use of resources, as reflected in the final growth yield of a population, is a selectable trait or not (Ibstedt et al, 2015; MacLean, 2008).”

“Under slow, absent or negative adaptation, clonal interference, positive epistasis, cell-cell interactions and death may all be substantial. In such evolutionary scenarios, more complex models may be needed”

We comment on the fitness estimation strategy in the response to reviewer 3 (comment 19) below.

7. I would like to see more profiles of adaptation in (much) smaller starting populations, to rule out the possibility of standing variation (adaptive mutations that are already present in the founder population) contributing to the adaptation.

Author response: The proposed design is conceptually interesting and straight-forward to perform. However, after several tries we conclude that it comes with unavoidable systematic bias attached. By substantially reducing the population size, we go much below the level at which our instruments can detect changes in population size. That early growth is therefore obscured in the first growth cycle. This results in an overestimation of the length of the lag phase and of the minimum doubling time (which tends to occur early). In the end, this leads to a false impression of adaptation between the first and the second cycle. We are not comfortable with including this artifact in the paper. However, we do believe that experiments with populations P5-P8, which are initiated from different colonies expanded from identical single cells, constitute compelling evidence against adaptation driven by standing variants. To increase confidence in this conclusion, we now address the problem analytically. The Results read:

“The extraordinarily fast As(III) adaptations could conceivably be due to a single rare adaptive mutation standing at substantial frequency in the shared founder population rather than *de novo* mutations. To account for this possibility, we repeated the arsenic adaptations in four new populations, hereafter termed P5-P8, that were initiated from distinct founder populations derived by clonal expansion from four different single cells (see Methods). As the original As(III) adapting populations (P1-P4), populations P5-P8 showed the same ultrafast and near deterministic adaptive leaps that with the higher sampling density were detectable already after 10 generations (Fig S2A). The probability of adaptive variants standing in all P5-P8 populations was estimated to be 3.9×10^{-5} (see Methods).”

The Methods read:

“The follow-up experiment of arsenic adapting populations, P5 to P8, was initiated identically, except that each of the four populations was initiated from four different founder populations. These were clonally expanded from four distinct cells, up to a population size of $\sim 3 \times 10^7$. Assuming that adaptive mutations during the clonal expansion from a single cell are Poisson distributed, with normal mutation rates and the mutation target sizes reported in Fig S6, the probability that a single P5-P8 population housed one or more standing adaptive variants is ~ 0.08 . The probability that all of P5-P8 housed one or more standing adaptive variant at experiment start is $\sim 3.9 \times 10^{-5}$.”

8. The model used for investigating the effect of epigenetics on fast emergence of beneficial mutations is not completely clear to me. As far as I understand, the beneficial mutation targets are limited to the genes identified. Ideally, should this not also include (unknown) targets? Can this be discussed a bit more in the text?

Author response: Unfortunately, we cannot provide estimates for, or reasonably model, the effect of the unidentified beneficial mutations individually. What we can do is estimate and model the aggregated effect of all unidentified mutations, as the difference between the performance of the population and that of clones carrying the reconstructed, single large effect mutation. This is reported in Fig S8. The relevant Results section reads:

“The reconstructed *FPS1*, *ASK10* and *ACR3* mutations explain most of the fitness gains seen in P1-P4, but not all of it. Unidentified causal mutations account for $\sim 25\%$ of total fitness gains, and this fraction is larger in *ASK10* and *ACR3* populations than in *FPS1* populations. We may therefore have underestimated the late fitness of *ACR3* and *ASK10* containing clones relative to *FPS1* containing clones. Indeed, when we mimicked this possibility by letting the adaptive gains of *ASK10* and *ACR3* mutations approach that of *FPS1*, *FPS1* was much less prone to fixate in *ASK10/ACR3* backgrounds (Fig S8).”

9. In the first paragraph of the results section, the authors say that the populations exposed to arsenic go "from poor to optimal performance" and they refer to figure 1A. However, that figure seems to indicate that the lag phase becomes longer as generations go by, which suggests that the populations

are not (yet) optimal, and that there may be a tradeoff. Many figures represent the doubling time, efficiency and lag time of the evolved/evolving populations compared to founder. This implies that the ratio of the evolved populations over the founder is used. In this case, adaptive changes in fitness would occur when the ratio for doubling time and lag time becomes negative; or when the ratio for efficiency becomes positive. However, all relative performances in Fig 1 increase, suggesting that the absolute doubling time increases and the lag time increased in the evolved populations. Please discuss. Moreover, I find the evolution experiments poorly described. What was the culture volume and system? Which was the initial OD? Did the cultures reach stationary phase before each transfer? These factors affect the selection regime and thus also the outcome of the experiment. The same questions apply to the experiments used to extract the fitness components.

Author response: We have now amended the Methods section. It reads:

“When comparisons across plates were made, $\log(2)$ estimates were first normalized to the corresponding mean of 4-20 WT (founder) controls distributed in fixed but randomized positions. For doubling time and lag, the relative growth measures equalled: mean of $\log(2)$ WT estimates - $\log(2)$ experimental estimate. For the population growth efficiency, the relative growth measure equalled: $\log(2)$ experimental estimate - mean of $\log(2)$ WT estimates. Positive values thus always indicate adaptation. The normalization accounts for systematic bias between plates, instruments and batches. Finally, a mean was formed across replicates.”

See response to question 7 below for further comments on the experimental design.

10. While the main text is well written, the figure legends and methods section are below standards. Specifically, the authors should describe in more detail how the experimental evolution was performed (temperature, were the cells grown in flasks or tubes or plates, in what volume, etc.). Other specific sentences that were unclear are listed in the minor comments.

Author response: The methods section has now been much expanded to better describe the experimental design. It reads:

“Experimental evolution: Except for the four follow-up arsenic adapting populations, P5-P8, reported in Fig S2A, all experimental evolutions were initiated from a single founder population. The founder population was constructed by clonal colony expansion up to an estimated 1 million cells, from a single cell, on SC agar medium (as above, + 2% (w/v) agar) with no added stress. The colony was dissolved in liquid SC medium to create the founder population, the optical density was measured, and an average of 10^5 cells were randomly drawn by pipetting of $5 \mu\text{L}$ of cell suspension into experimental wells to initiate each adaptation. The follow-up experiment of arsenic adapting populations, P5 to P8, was initiated identically, except that each of the four populations was initiated from four different founder populations. These were clonally expanded from four distinct cells, up to a population size of $\sim 3 \times 10^7$. Assuming that adaptive mutations during the clonal expansion from a single cell are Poisson distributed, with normal mutation rates and the mutation target sizes reported in Fig S6, the probability that a single P5-P8 population housed one or more standing adaptive variants is ~ 0.08 . The probability that all of P5-P8 housed one or more standing adaptive variant at experiment start is $\sim 3.9 \times 10^{-5}$. Experimental evolutions were performed in a batch-to-batch mode in flat bottom 96-well micro-titre plates containing SC complete medium supplemented with stress factors (Table S1). To reduce the risk of cross-contamination, every second well were left empty, such that all pairs of populations were separated by empty wells. No indication of cross-contamination between As(III) populations were found in the sequence data. Except for the follow-up As(III) adapting populations P5-P8, populations were propagated over 50 batch-to-batch cycles as $175 \mu\text{L}$, non-shaken cultures maintained at 30°C . The follow-up As(III) adapting populations P5-P8 were propagated over 20 cycles. In all cycles, populations were cultivated well into stationary phase. The cultivation length corresponding to $\sim 120\text{h}$ cultures over the first 10 cycles, $\sim 96\text{h}$ in cycles 10-30 and $\sim 72\text{h}$ in cycles 30 to 50. Stationary phase population sizes corresponded to on average $N = 3.5 \times 10^6$ cells, with the largest deviation corresponding to half that size. To initiate each new cycle, $5 \mu\text{L}$ of re-suspended and randomly drawn stationary phase cell cultures, corresponding to an average of $N = 10^5$ cells, were multi-pipetted into fresh medium. Each batch cycle

corresponded to ~5 population size doublings. The adaptation schema thus progressed over ~250 population doublings (~100 doublings for follow-up As(III) adapting populations P5-P8). Except for the follow-up arsenic adaptations, P5-P8, 50 μL of each population was sampled at the end of every 5th cycle, pipetted into 100 μL of 30% glycerol and stored as a frozen fossil record at -80 C. For populations P5 to P8, sampling was instead performed at every batch cycle.

Fitness component extraction: To estimate fitness components, frozen samples were first thawed and re-suspended. 10 μL were pipetted into random wells in 100 well honey-comb plates, each well containing 350 μL of liquid SC medium. Populations were pre-cultivated without shaking at 30C for 72h until well into stationary phase. Following re-suspension, 10 μL of each pre-culture was randomly sampled and transferred to 100 well honey-comb plates, containing 350 μL of liquid SC medium supplemented by relevant stress factors. Populations were cultivated in Bioscreen C (Growth curves Oy, Finland) instruments for 72h at 30C and at maximum horizontal shaking for 60s every other minute (Warringer & Blomberg, 2003; Warringer et al, 2003). Optical density (turbidity) was recorded every 20 min using a wideband (420-580nm) filter. Stochastic noise was removed by light smoothing of the raw data, the background light scattering was subtracted and optical densities were transformed into population size estimates using an empirically based calibration (Fernandez-Ricaud et al, 2016). From population size growth curves, population doubling times, length of the lag phase and population growth efficiency (total gain in population size) were extracted (Fernandez-Ricaud et al, 2016).”

Minor questions and suggestions

11. While growth rates and doubling times are two sides of the same coin ($GR = \ln(2)/DT$), they should not be used interchangeably. When presenting doubling times, this should not be labeled as rates. This is especially confusing in Fig 2b, which shows that the 'rate' of the founder is higher in the presence of arsenic (compared to a benign condition), suggesting that the founder is growing at a higher growth rate in the presence of arsenic. It is also difficult to understand why a $\log(2)$ transformation is necessary when presenting these absolute measures of doubling time, efficiency and lag.

Author response: We agree and have changed all labels and text statements to doubling times. We prefer to keep the log transformation of data: it is standard in omics science as it makes normal distribution assumptions less precarious. The Discussion now reads:

“Population growth parameters were $\log(2)$ transformed to better adhere to normal distribution assumptions.”

12. Second paragraph of Results section: which are the "repeated arsenic adaptations P5-P8"? Populations P1-P4 and P5-P8 are never explained.

Author response: We now explain the distinction between populations P1-P4 and P5-P8 in detail in the Methods sections. See response to comment 7. Also, the Results now read:

“The four populations, hereafter termed P1-P4, exposed to arsenic in its most toxic form, As(III), adapted faster than populations exposed to other challenges”

13. Page 6, "positive pleiotropy vastly accelerated adaptation": according to the figures, model 2 and model 3 don't seem to be so vastly different.

Author response: We have now amended this sentence. The Results reads:

“Populations exclusively experiencing positive pleiotropy adapted only moderately faster than populations experiencing both positive and negative pleiotropy. Thus, positive pleiotropy between fitness components can indeed accelerate adaptation drastically, and the presence of negative pleiotropy only moderately limits this acceleration.”

14. Last paragraph in page 6: the authors say they assume that the mutations emerge early. If they would sequence the intermediate time-points, they could know for sure when each of the mutations emerge.

Author response: We have now sequenced intermediate time points and confirm that the mutations emerge and fix early (see response to reviewer 1, comment 1).

15. Page 7, "adaptive speed did not accord with experimental data": Where can we see this?

Author response: To clarify, we have amended this section slightly. It now reads:

“While the predicted large variations in adaptive speed between populations at a basal mutation rate (Fig 3D - upper left panel; compare time to vertical black line) is a possible scenario, it produces a distribution of simulated adaptations from which it is unlikely to draw the four nearly deterministic ultrafast adaptations observed in the experimental data (Fig 1B, Fig S2A). A mutation rate closer to the upper bound of empirical estimates of the basal mutation rate (3x) (Lang & Murray, 2008; Lynch, 2006; Lynch et al, 2008) increased the homogeneity in adaptive speed considerably, while allowing heterogeneity in adaptive solutions. In this case, the founder genotype went extinct in 35 generations in the median population (Fig 3D, upper right panel). Mutation rates (>5x) above empirical estimates of the basal mutation rate gave results that were incompatible with the experimental data, as the superior *FPSI* mutations were then consistently fixed, leaving no room for *ACR3* and *ASK10* based solutions (Fig 3D, lower panel). The simulations therefore suggested a mutation rate between 1x and 3x at loci under selection, and excluded substantially higher mutation rates.”

Reviewer #3: The paper by Gjuvsland et al attempts to argue that the pattern of evolution observed in a adaptation of yeast to high arsenic is fully compatible with the absence of epigenetic mechanism that increase the rate of mutation. In this way, the authors, claim they can disentangle the genetic and epigenetic mechanisms. I do not believe this paper is ready for publication in any journal, and certainly not in Mol. Syst. Biology.

16. The description of the experiment is obscure with most results in Supp. Information.

Author response: We have now substantially expanded the Methods section. See response to reviewer 2, comment 10.

17. The first paragraph of the Results starts with citing Table S1 and Fig. S1A, S1B, C, and D.

Author response: We have now promoted Fig S1A and S1D to actual figures (Fig 1A and Fig1D respectively.)

18. It is not at all obvious how the fitnesses and clonal interference patterns have been characterized and from what I could surmise with great difficulty consisted of just sequencing a few clones.

Author response: We now better describe the fitness component extraction and make clear that the sequencing is population sequencing, not clone sequencing (see response to reviewer 2, comment 10). Clonal interference is not expected to be a major evolutionary phenomenon under the current ultrafast adaptation scenario as a single very strong mutation rapidly emerges and drive to fixation. Indeed, this is now shown empirically in the new Fig 2B. It is also reflected in the simulations shown in Fig 3C and 3D.

19. The paper brings very little new thinking to the field as well. Desai and Fisher 2007 and subsequent work is an attempt to explain the data using simple models and population sequencing and even more so barcoding experiments of Levy et al provide much better resolution to answer these questions.

Author: We respectfully disagree. We are not aware of any previous experimental-theoretical attempts to distinguish the roles of genetics and non-genetics in the fastest adaptation scenarios. Moreover, whereas the work by Desai et al, now cited in the Discussion, does combine experimental and theoretical population genetics perspectives, its main strength is on explaining steady state

adaptation where a population has evolved in a constant environment for a long time such that the distribution of fitness effects is stable. It does not break down fitness into its component, and thus does not allow for exploring the causality underlying fitness effects. The Levy barcoding technique provides excellent resolution when estimating the distribution of selection coefficients in adapting populations. However, it does not directly estimate adaptation and does not provide an evident way forward for understanding why a particular adaptation pattern emerges in a particular environment. We summarize our view of this in the Discussion which now reads:

“There are currently two approaches to measure and model fitness in experimental populations (Barrick & Lenski, 2013). The standard approach measures the fitness of individual genotypes as their frequency change over time in competition assays (Gresham et al, 2008; Lang et al, 2011). This is simplified if each genome in a population is barcoded before the onset of selection with a unique sequence tag (Levy et al, 2015; Venkataram et al, 2016), allowing very accurate estimation of the fitness distribution of standing and de novo mutations for use as a model input. Given that change in fitness of the population is also exactly measured and a suitable modelling framework in place, such approaches are certainly useful for understanding the speed of adaptation. So far however, these approaches have focused on steady state adaptation where a population has evolved in a constant environment for a long time, with selection acting only on doubling time (Kosheleva & Desai, 2013; Rice et al, 2015).

We employed the alternative approach: break fitness in batch-to-batch experiments down into its components, both experimentally and theoretically. This approach certainly comes with caveats attached. It is not always clear that the estimated and modelled fitness components - here cell division time and time to the first cell division - fully captures fitness. In experimental microbial populations, death rates may not be negligible and it is debatable whether efficient use of resources, as reflected in the final growth yield of a population, is a selectable trait or not (Ibstedt et al, 2015; MacLean, 2008). Furthermore, to estimate fitness components, mutations must be reconstructed or reversed and the fitness component of individual genotypes must be estimated. This is laborious, in particular if interactions between mutations and between individuals (Moore et al, 2013) are to be measured. Here, we considered evolutionary scenarios of very fast adaptation, where single mutations drive adaptation and rapidly rise to fixation, without measurable death occurring. In such scenarios, the caveats above are lesser concerns. Under slow, absent adaptation or negative adaptation (extinction), clonal interference, epistasis, genetic hitchhiking, cell-cell interactions and death may all be substantial. In such evolutionary scenarios, more complex models may be needed.

A marked benefit of breaking fitness down, and connecting it to genotypes via the intervening phenotypic layers, is the possibility to identify the causal factors underlying particular patterns of adaptation. This is illustrated by our discovery that positive pleiotropy between fitness components is the driving force of the observed ultrafast adaptation. To understand adaptation dynamics at an even deeper level, both experimentation and modelling must be extended to molecular phenotypes. For example, by connecting the time to the first cell division and the cell division time to the biochemical and network properties of As(III) metabolism (Talemi et al, 2014), a complete and formalized understanding of the causes of ultrafast As(III) adaptation could be obtained.”

20. Finally, the fit of poor data into a simple model is a poor way to argue that epigenetic mechanisms could not have been involved.

Author response: It is unclear to us why the reviewer considers the data to be poor. Fitness components are estimated with very high accuracy using a randomized, high replication experimental design and analytically connected to fitness as cell division time and time to the first cell division. We now also show that death rates are negligible (see response to reviewer 2, comment 6). Given the fast adaptation, other confounding factors are few, as now discussed in the discussion (see response to comment 19). All models are simplifications of reality, because of computational reasons, the absence of estimates for some parameters, and ease of interpretation of outcomes. We mostly employ an individual based model, in which each cell is equipped with its own genotype-phenotype map. This is as realistic as population genetics modelling can be and computationally very intensive. Whereas we do disregard some parameters for which relevant

estimates are not available, such as cell-cell interactions and positive epistasis, these are expected to be of little relevance in the ultrafast scenarios considered here. This now discussed in the Discussion (see response to comment 19). Finally, we are not aware of any alternative approach capable of evaluating the transient contribution of epigenetics, e.g. in the form of local elevations of the mutation rate, to adaptation dynamics. This is a serious caveat in discussions concerning the explanatory power of the neo-Darwinistic paradigm.

2nd Editorial Decision

27 October 2016

Thank you again for submitting your work to Molecular Systems Biology. We have now heard back from the two referees who accepted to evaluate the revised study. As you will see, the referees find the topic of your study of potential interest and are supportive and I am please to inform you that we will be able to accept your paper for publication in Molecular Systems Biology pending the following minor points:

- Supplementary files should be combined into an 'Appendix' that starts with a Table of Content. Please update the call-outs to 'Appendix Fig S1', 'Appendix Table S1', etc, *both in Appendix figure legends and in main text*.
- We are grateful that you deposited your code on bitbucket. However, for long term archival purpose, we would kindly ask you to also include the zipped archive of the code as "Computational model EV1" and include the respective call out from the Data and model availability section.
- We would also kindly ask you to deposit the sequencing data in an appropriate public repository (see our Guide to Authors) and include the respective accession number in the Data and model availability section.

REFEREE REPORTS

Reviewer #1:

The authors have addressed my concerns. I disagree with referee 3 - that genetic adaptation is fast enough to account for the behaviour - by Occam's razor - make the involvement of epigenetic mechanisms unlikely.

Reviewer #2:

The authors have carried out several new experiments to address my major concerns (or at least acknowledge some of the possible caveats that I identified in the discussion). So, as far as I am concerned, I am OK with proceeding to publish.

2nd Revision - authors' response

11 November 2016

Editorial requests:

Thank you again for submitting your work to Molecular Systems Biology. We have now heard back from the two referees who accepted to evaluate the revised study. As you will see, the referees find the topic of your study of potential interest and are supportive and I am please to inform you that we will be able to accept your paper for publication in Molecular Systems Biology pending the following minor points:

- Supplementary files should be combined into an 'Appendix' that starts with a Table of Content. Please update the call-outs to 'Appendix Fig S1', 'Appendix Table S1', etc, *both in Appendix figure legends and in main text*.

Authors: Corrected as requested.

- We are grateful that you deposited your code on bitbucket. However, for long term archival purpose, we would kindly ask you to also include the zipped archive of the code as "Computational model EV1" and include the respective call out from the Data and model availability section.

Authors: Corrected as requested.

The text reads: "Models are available as Data Model EV1 and can also be downloaded from https://bitbucket.org/ajkarloss/yeast_sim."

- We would also kindly ask you to deposit the sequencing data in an appropriate public repository (see our Guide to Authors) and include the respective accession number in the Data and model availability section.

Authors: Corrected as requested.

The text reads: "The SOLiD sequencing data is accessible at EBI (<http://www.ebi.ac.uk/ena/data/view/PRJEB17740>) with accession number PRJEB17740. The Illumina sequencing data is accessible at NCBI (<https://www.ncbi.nlm.nih.gov/sra?term=SRP092403>) with accession number SRP092403."

Reviewer #1:

The authors have addressed my concerns. I disagree with referee 3 - that genetic adaptation is fast enough to account for the behaviour - by Occam's razor - make the involvement of epigenetic mechanisms unlikely.

Reviewer #2:

The authors have carried out several new experiments to address my major concerns (or at least acknowledge some of the possible caveats that I identified in the discussion). So, as far as I am concerned, I am OK with proceeding to publish.

3rd Editorial Decision

16 November 2016

Thank you again for sending us your revised manuscript. We are now satisfied with the modifications made and I am pleased to inform you that your paper has been accepted for publication.

Corresponding Author Name: Arne Gjuvsland, Jonas Warringer

Manuscript Number: MSB-16-6951